# ContextIF: Enhancing Instruction-Following through Context Reward

**Yule Zhong    Jiacheng Yao    Guoxiu He**[*]

School of Economics and Management, East China Normal University, Shanghai, China
yulez@stu.ecnu.edu.cn    jcyao@stu.ecnu.edu.cn    gxhe@fem.ecnu.edu.cn

## Abstract

While supervised fine-tuning (SFT) and preference learning (PL) are widely used to enhance the instruction-following ability of Large Language Models (LLMs), they often struggle to generalize to novel or complex instructions and may compromise the models' general capabilities. In-Context Learning (ICL) emerges as a promising alternative due to its strong generalization without modifying the model's parameters, but its effectiveness is constrained by the reliance on high-quality, manually curated demonstration pools. To overcome this limitation, we propose ContextIF, a reinforcement learning (RL) framework for automatic context generation. Guided by comprehensive context reward, ContextIF is optimized by Group Relative Policy Optimization (GRPO). It aims to generate precise constraint summaries and optimal context demonstrations tailored to given instructions, thereby improving the instruction-following performance of target LLMs. We evaluate ContextIF on multiple representative instruction-following benchmarks using popular open-source LLMs. Experimental results demonstrate that ContextIF achieves substantial performance gains over existing SFT and ICL methods, while also generalizing effectively to unseen constraint conditions. Moreover, ContextIF preserves the parameters and general capabilities of the target models, offering strong adaptability and scalability. Our code is available at https://github.com/ECNU-Text-Computing/ContextIF.

## 1 Introduction

Large Language Models (LLMs) have achieved remarkable performance across diverse domains in natural language processing (NLP) (Touvron et al., 2023; Achiam et al., 2023; GLM et al., 2024). As these models are increasingly deployed to develop agents across diverse domains, effective instruction-following has become a critical factor for their practical application. Agents must comply with various constraints and instructions to ensure safe, trustworthy, and reliable interactions (Li et al., 2024; Tu et al., 2024; He et al., 2025a). However, existing LLMs often struggle to adhere to the complex, multi-faceted constraints common in real-world instructions, limiting their effectiveness (Zhou et al., 2023; Sun et al., 2024; Qin et al., 2024; Xia et al., 2024). Although numerous methods have been proposed to improve instruction-following capabilities, most studies focus on pre-training, supervised fine-tuning (SFT), preference learning (PL) and reinforcement learning from human feedback (RLHF). These approaches require substantial high-quality data and computational resources, and are susceptible to catastrophic forgetting, which inevitably undermines previously acquired knowledge and generalization performance (Lin et al., 2023; Song et al., 2025). Moreover, they face considerable challenges in generalizing to unseen constrained tasks.

In-Context Learning (ICL) is a capability that enables LLMs to learn directly from demonstrations provided within the input prompt (Brown et al., 2020). By leveraging just a few task-specific demonstrations, LLMs can achieve strong performance across diverse tasks and rapidly adapt to new domains or problems without extensive fine-tuning (Yao et al., 2025). This capability is particularly crucial for enhancing instruction-following in scenarios involving complex, multi-constrained requirements. Existing research has explored the importance of ICL in instruction-following tasks and investigated the impact of context (Zeng et al., 2025; Zhao et al., 2024; Li et al., 2025). Their

---

[*]Corresponding author

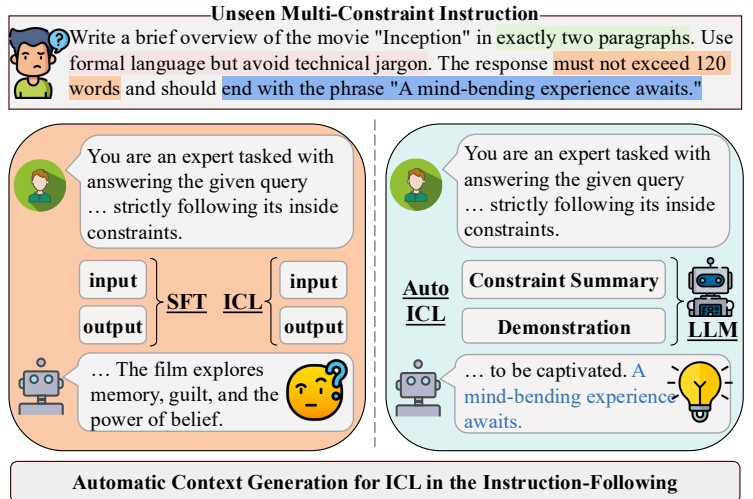

Figure 1: Comparison of conventional SFT/ICL methods (Left) with our proposed ContextIF (Right). Traditional SFT and ICL rely on extensive, high-quality human-annotated datasets, struggling to generalize to unseen constraints. In contrast, ContextIF enhances instruction-following performance by automatically generating high-quality constraint summaries and demonstrations.

findings indicate that ICL performance is highly contingent upon the quality of the provided examples, as different demonstrations can significantly affect the model's final adherence, underscoring the critical importance of context optimization for this task.

While promising, the efficacy of ICL is tethered to the quality and relevance of the provided demonstrations. Prior research has predominantly approached this challenge through two paradigms: manual curation and retrieval-based selection (Wang et al., 2024). Manual curation, though capable of producing high-quality examples, is expensive and fails to scale to diverse real-world instructions. Retrieval-based methods offer better scalability but are constrained by the coverage and quality of the source pool. This dependency becomes a critical bottleneck for complex instruction-following, as illustrated in Figure 1; user instructions often contain nuanced or novel constraints for which suitable demonstrations do not exist in static datasets. To address this, recent studies propose that automatically generating task-specific context via LLMs is key to unlocking the full potential of ICL (Chen et al., 2023; Lee et al., 2025). However, self-generation without quality verification is often unreliable, resulting in demonstrations that lack the structural rigor and semantic alignment required for effectiveness. Our work is motivated by the need for a principled framework to reliably generate adaptive, task-specific context to enhance instruction-following.

In this paper, we propose ContextIF, a novel framework that empowers LLMs to autonomously generate their own optimal context for instruction-following. Deviating from the traditional reliance on static pools, ContextIF leverages a generator model to synthesize high-quality, task-specific context tailored to any given user query. The core of our framework is a reinforcement learning (RL) pipeline designed to optimize this generation process. The workflow follows a two-stage strategy: first, deconstructing the user query into a concise constraint summary; and second, constructing a parallel demonstration that exemplifies these deconstructed constraints. To guide this intricate process, we introduce the context reward, a comprehensive composite signal that evaluates both structural rigor and semantic alignment. The entire RL process is stabilized and optimized using Group Relative Policy Optimization (GRPO) (Shao et al., 2024), enabling the generator model to learn how to produce context that maximally enhances instruction-following performance.

We conduct extensive experiments across two leading open-source models: LLaMA3-8B-Instruct and Mistral-7B-Instruct. To rigorously assess instruction-following, we utilize a comprehensive suite of benchmarks, including IFEval, Multi-IF, FollowBench, and LiveBench. Our findings demonstrate that backbone LLMs equipped with ContextIF consistently and significantly outperform strong baselines, including traditional SFT methods and advanced ICL strategies, even those leveraging demonstrations from manually curated pools or teacher models as powerful as GPT-4o.

Furthermore, our analysis reveals two critical advantages of our RL-based approach. First, ContextIF excels at generalizing to unseen constraint types. Second, it preserves and even enhances the model's foundational capabilities, directly addressing the common challenge of catastrophic forgetting associated with fine-tuning methods.

Our main contributions are summarized as follows:

• We introduce ContextIF, a novel RL-based framework that empowers LLMs to autonomously generate their own optimal context. This approach overcomes the static data limitations of traditional methods and enables adaptive demonstration generation tailored to any given instruction.

• We design a multi-faceted Context Reward that provides precise guidance for the RL process. This reward mechanism effectively quantifies both the structural rigor and semantic alignment of generated contexts, ensuring high-quality and task-relevant demonstrations.

• Extensive experiments show that ContextIF sets a new state-of-the-art for instruction-following among open-source models. Furthermore, our approach exhibits superior generalization to unseen constraints and effectively mitigates catastrophic forgetting, thereby preserving the model's foundational capabilities.

## 2 RELATED WORK

We review two lines of related work: instruction-following methods and In-Context augmentation.

### 2.1 INSTRUCTION-FOLLOWING

Instruction-following is a fundamental capability of LLMs, requiring them to understand and generate responses that adhere to complex human instructions (Li et al., 2023b; Dong et al., 2024). Recent research has primarily focused on constraints within instructions, such as keywords and length (Zhou et al., 2023). To assess these capabilities, numerous benchmarks have been developed, ranging from those based on synthetic instructions and rule-based evaluations (Zhou et al., 2023; Yao et al., 2023; Iso, 2022) to those utilizing complex datasets and model-based scoring (Jiang et al., 2024; Qin et al., 2024; Wen et al., 2024). Based on these benchmarks, various methods have been proposed to enhance instruction-following, mainly focusing on (1) SFT data scaling (Sun et al., 2024; Dong et al., 2024; Ren et al., 2025), which involves collecting high-quality fine-tuning data through model distillation, back-translation, or iterative response refinement (An et al., 2025); (2) preference data curation, such as code verification (Dong et al., 2024) and model validation (An et al., 2025; Cheng et al., 2024); and (3) reinforcement learning with reward verification (Peng et al., 2025; Lambert et al., 2024), incorporating verification-based reward signals to steer model behavior. However, conventional approaches to improving instruction-following have centered on data collection and targeted training. This methodology is prone to several critical issues, including catastrophic forgetting of prior knowledge and a degradation of generalist abilities. Moreover, such models typically exhibit poor generalization when faced with novel constraints.

### 2.2 IN-CONTEXT AUGMENTATION

The ICL paradigm enables LLMs to perform novel tasks without parameter updates by leveraging task-specific input-output pairs, known as demonstrations (He et al., 2025b; Moeini et al., 2025). In many instruction-following scenarios, ICL has proven to be a highly effective alternative to fine-tuning, allowing models to adapt to new requirements with minimal data (Zhao et al., 2024; Zhang et al., 2025). Existing research on ICL optimization has primarily focused on the selection and ordering of demonstrations from pre-defined, static pools (Wang et al., 2024; Dherin et al., 2025). Given the cost of manually crafting prompts, recent studies have shifted toward automated prompt generation (Chen et al., 2023). However, these approaches still face significant bottlenecks: retrieval-based methods are limited by the coverage and quality of their source pools (Li et al., 2023a; 2025), while self-generation methods often produce demonstrations that lack structural rigor and semantic alignment with the user's specific constraints. To address these limitations, we propose to leverage RL to optimize the context generation process. Unlike previous methods, our approach employs a principled reward mechanism to guide the synthesis of demonstrations, ensuring high-precision context generation and superior generalization to unseen tasks.

## 3 METHODOLOGY

### 3.1 OVERVIEW OF CONTEXTIF

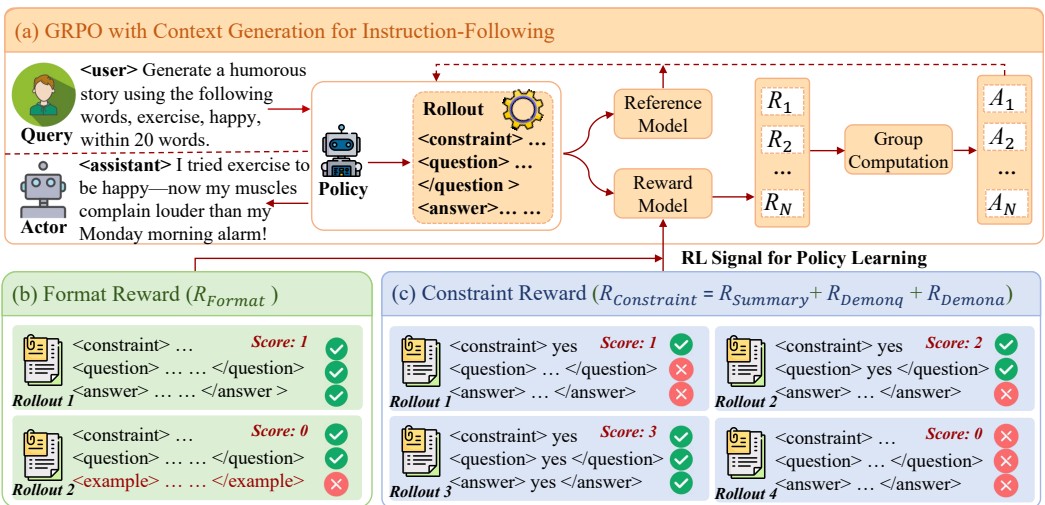

Figure 2: An overview of the ContextIF framework. (a) The policy model, trained with GRPO, generates a constraint and demonstration context block based on a user query. This output is then evaluated by reward model to compute the final RL signal. (b) The Format Reward provides a binary signal for structural correctness. (c) The Constraint Reward provides a fine-grained score based on the semantic quality of the summary and the demonstration, guiding the policy toward generating task-optimal context for instruction-following.

We introduce ContextIF, a RL framework designed to autonomously optimize context generation for instruction-following. ContextIF aims to overcome the limitations of both static, manually curated demonstration pools and context naively generated by LLMs without quality verification. As illustrated in Figure 2, ContextIF employs a policy model to act as the context generator, which is initialized from the same base LLM as the actor model. Given a user query, the generator's task is to synthesize a structured ICL demonstration, comprising a precise constraint summary and a corresponding parallel question-answer pair. This entire process is optimized via a context reward signal, which guides the generator to produce context that is both structurally sound and semantically effective for the target instruction-following task.

### 3.2 CONTEXT ROLLOUT

The primary objective of the context rollout phase is to autonomously synthesize a task-specific context tailored for the input query. As illustrated in Figure 2(a), the policy model receives the user query and, guided by a specialized system prompt, performs an analytical generation step to construct the augmented context rather than directly resolving the query.

Specifically, the policy executes a single rollout to produce a structured XML block. This block is designed to be self-contained, comprising three functional tags: <constraint>, <question>, and <answer>. The <constraint> tag encapsulates the model's deconstruction of the original instruction's requirements. Subsequently, the <question> and <answer> tags jointly form a parallel demonstration that instantiates these identified constraints in a new, analogous scenario. Together, these components provide all the necessary context for the model to adhere to complex instructions without external data.

This generated context block constitutes the complete trajectory for the RL process. Unlike conventional ICL, which relies on demonstrations manually selected from a static pool, ContextIF dynamically crafts a bespoke context for each individual query. As depicted in Figure 2(a), this synthesized context is evaluated by a multi-dimensional reward model. The resulting reward signal, which quantifies the quality and effectiveness of the generated context, is then used to update

the policy via group-based relative computation. This iterative loop trains the policy to become an expert context generator, capable of producing optimal augmentations that significantly bolster instruction-following performance. Detailed prompts for model inference and the reward evaluation process are provided in Appendix D.1.

## 3.3 REWARD DESIGN

Verifiable reward signals have demonstrated remarkable empirical success, emerging as a pivotal technique for aligning LLMs with complex requirements. In training ContextIF, we employ a composite reward function that synergizes structural verification with semantic assessment. Specifically, the format reward evaluates whether the model's output strictly conforms to the prescribed XML schema, while the constraint reward quantifies the fidelity of the deconstructed constraint summary and the synthesized demonstration. Formally, the total reward $\mathcal{R}_{\text{context}}$ is decomposed into these two primary components: $\mathcal{R}_{\text{format}}$ and $\mathcal{R}_{\text{constraint}}$, which are detailed as follows.

**Format Reward.** The format reward $\mathcal{R}_{\text{format}} \in \{0, 1\}$ evaluates whether the model output strictly follows the prescribed XML schema. It verifies the presence of the constraint summary, demonstration question, and demonstration answer tags, ensuring they appear in the exact sequence specified by the task requirements:

$$\mathcal{R}_{\text{format}} = \begin{cases} 1, & \text{if all required tags appear in the correct order;} \\ 0, & \text{otherwise.} \end{cases} \tag{1}$$

**Constraint Reward.** The constraint reward $\mathcal{R}_{\text{constraint}} \in \{0, 1, 2, 3\}$ evaluates the semantic integrity of the generated context. We employ a strong LLM (*e.g.*, LLaMA3-70B-Instruct) as a programmatic judge to assess generated context based on three binary criteria:

- **Summary Accuracy ($r_{\text{sum}}$):** Awarded 1 if the <constraint> tag accurately encapsulates all constraints derived from the original query, and 0 otherwise.
- **Question Parallelism ($r_{\text{demoq}}$):** Awarded 1 if the generated <question>, serving as the *demonstration query*, instantiates a parallel constraint structure analogous to the user's input, and 0 otherwise.
- **Constraint Adherence ($r_{\text{demoa}}$):** Awarded 1 if the generated <answer>, acting as the *demonstration response*, faithfully satisfies all specific constraints defined within the synthesized question, and 0 otherwise.

The total constraint reward is defined as the unweighted sum of these components:

$$\mathcal{R}_{\text{constraint}} = r_{\text{sum}} + r_{\text{demoq}} + r_{\text{demoa}}. \tag{2}$$

The final reward $\mathcal{R}_{\text{context}}$ is derived as the aggregate of the structural and semantic components:

$$\mathcal{R}_{\text{context}} = \mathcal{R}_{\text{format}} + \mathcal{R}_{\text{constraint}}. \tag{3}$$

In summary, our reward design explicitly decouples structural adherence from semantic fidelity. By unifying strict XML compliance with a fine-grained evaluation of the synthesized context, the composite reward signal guides the policy to produce outputs that are not only syntactically robust but also contextually optimal for complex instruction-following. This holistic evaluation framework is essential for transforming the policy into a proficient context generator, ultimately ensuring the effectiveness of the dynamically generated context in downstream instruction-following scenarios.

## 3.4 RL TRAINING WITH GRPO

To optimize the policy model for high-quality context generation, we adopt GRPO (Shao et al., 2024). Unlike traditional actor-critic methods, GRPO estimates a relative baseline within a group of samples, making it highly efficient for training LLMs on structured output tasks. For each query $q$ sampled from the dataset $\mathcal{D}$, the policy $\pi_\theta$ generates a group of $G$ independent context rollouts $\{c_1, \dots, c_G\}$.

**Group-Relative Advantage.** Each rollout $c_i$ is evaluated by the reward model to obtain a composite reward $r_i$, which integrates structural and semantic signals:

$$r_i = \mathcal{R}_{\text{format}}(c_i) + \mathcal{R}_{\text{constraint}}(c_i). \tag{4}$$

To estimate the advantage without a separate critic, we compute the mean $\mu_G$ and standard deviation $\sigma_G$ of the rewards within each group:

$$\mu_G = \frac{1}{G} \sum_{i=1}^{G} r_i, \quad \sigma_G = \sqrt{\frac{1}{G} \sum_{i=1}^{G} (r_i - \mu_G)^2}. \tag{5}$$

The normalized advantage $\hat{A}_i$ for each rollout is then derived as:

$$\hat{A}_i = \frac{r_i - \mu_G}{\sigma_G + \epsilon}, \tag{6}$$

where $\epsilon$ is a small constant for numerical stability. This formulation encourages the policy to prioritize context structures that exhibit relatively superior quality within the same task distribution.

**Optimization Objective.** The policy $\pi_\theta$ is updated by maximizing the expected relative advantage while maintaining stability through a KL divergence constraint. Following the standard GRPO formulation, the objective function is defined as:

$$\mathcal{L}_{\text{GRPO}}(\theta) = \mathbb{E}_{q \sim \mathcal{D}, \{c_i\} \sim \pi_{\theta_{\text{old}}}} \left[ \frac{1}{G} \sum_{i=1}^{G} \left( \min \left( \rho_i(\theta)\hat{A}_i, \text{clip}(\rho_i(\theta), 1 - \alpha, 1 + \alpha)\hat{A}_i \right) - \beta D_{KL}(\pi_\theta \| \pi_{\text{ref}}) \right) \right], \tag{7}$$

where $\rho_i(\theta) = \frac{\pi_\theta(c_i|q)}{\pi_{\theta_{\text{old}}}(c_i|q)}$ represents the importance sampling ratio, $\alpha$ is the clipping threshold, and $\beta$ is the KL penalty coefficient. This holistic training objective guides the model to autonomously evolve into a proficient context generator, ensuring that the synthesized augmentations are both syntactically robust and contextually optimal for bridging the gap between complex instructions and the model's execution capabilities in downstream instruction-following scenarios.

# 4 EXPERIMENT

## 4.1 BASELINES

In our experiments, we compare ContextIF with a series of strong baselines specifically optimized for instruction-following, including Conifer (Sun et al., 2024), AutoIF (Dong et al., 2024), SPAR (Cheng et al., 2024) and UltraIF (An et al., 2025), which employs SFT and Direct Preference Optimization (DPO). We also include TULU 3 (Lambert et al., 2024), a representative open-source suite featuring advanced multi-stage post-training. Additionally, we evaluated various industrial models, including GPT-4o (Hurst et al., 2024) and QwQ-32B (Qwen, 2025). A detailed description of each baseline is provided in Appendix B.

## 4.2 BACKBONE LLMS

We evaluate ContextIF across two representative LLM families to ensure architectural diversity: (1) **LLaMA3-8B-Instruct** (Dubey et al., 2024), noted for its robust 15T-token pre-training and superior instruction-following baseline; (2) **Mistral-7B-Instruct** (Jiang et al., 2023), a widely adopted benchmark model utilizing Grouped-Query and Sliding Window Attention for efficient inference.

## 4.3 EVALUATION BENCHMARKS

To provide a holistic assessment of ContextIF, we evaluate our models on two distinct categories of benchmarks:

**Instruction-Following Benchmarks.** We select four representative benchmarks to assess adherence to complex constraints: **IFEval** (Zhou et al., 2023), the standard for verifiable formatting and instruction-following; **Multi-IF** (He et al., 2024), focusing on multi-turn and multilingual scenarios; **FollowBench** (Jiang et al., 2024), a comprehensive testbed for diverse constraint types; and **LiveBench** (White et al., 2024), a dynamic benchmark designed to mitigate data contamination.

Table 1: Evaluation results of different models on IFEval, Multi-IF, FollowBench (SSR), and LiveBench datasets. **P** and **I** stand for **P**rompt and **I**nstruction levels, respectively. **S** and **L** represent **S**trict and **L**oose metrics for IFEval. For LiveBench, we only report the performance on the subset of instruction-following data.

| Model | IFEval | | | | | Multi-IF | | | FollowBench | LiveBench |
|---|---|---|---|---|---|---|---|---|---|---|
| | P (L) | I (L) | P (S) | I (S) | Avg. | Turn1 | Turn2 | Turn3 | SSR | Score |
| GPT-4o | 84.80 | 89.60 | 79.90 | 85.60 | 84.98 | 82.30 | 71.70 | 59.30 | 75.30 | 64.90 |
| QwQ-32B | 86.10 | 90.40 | 82.80 | 88.00 | 86.83 | 64.20 | 56.60 | 48.40 | 73.50 | 62.20 |
| TULU 3 | 82.80 | 87.50 | 79.70 | 85.10 | 83.78 | 82.10 | 63.20 | 51.20 | 70.30 | 60.30 |
| LLaMA3-70B-Instruct | 84.04 | 89.21 | 77.76 | 84.53 | 83.89 | 81.90 | 65.80 | 53.90 | 70.90 | 61.70 |
| *LLaMA3-8B-Instruct Models* | | | | | | | | | | |
| LLaMA3-8B-Instruct | 77.02 | 84.05 | 69.44 | 77.94 | 77.11 | 63.83 | 52.24 | 43.92 | 62.90 | 46.70 |
| Conifer-8B | 79.50 | 85.50 | 75.60 | 82.70 | 80.83 | 66.00 | 53.80 | 41.90 | 64.76 | 46.90 |
| UltraIF-8B | 82.15 | 86.40 | 77.85 | 78.40 | 81.20 | 68.52 | 56.41 | 44.84 | 65.10 | 47.50 |
| AutoIF-8B | 81.25 | 85.50 | 76.45 | 78.16 | 80.34 | 65.55 | 53.92 | 43.11 | 63.80 | 47.10 |
| SPAR-8B | 81.15 | 87.05 | 79.11 | 85.13 | 83.11 | 72.55 | 60.46 | 51.32 | 68.80 | 49.80 |
| ContextIF-8B | 83.54 | 88.72 | 77.07 | 84.05 | **83.35** | **74.32** | **62.58** | **53.51** | **69.37** | **59.90** |

**General Capabilities.** To monitor potential alignment tax, we evaluate core competencies using: **MMLU** (Hendrycks et al., 2021) for general knowledge; **GSM8K** (Cobbe et al., 2021) for mathematical reasoning; **BBH** (Suzgun et al., 2023) for logical tasks; and **HumanEval** (Chen et al., 2021) for code generation.

Detailed descriptions of each benchmark and their corresponding evaluation protocols are provided in Appendix C.

## 4.4 MAIN RESULTS

All experimental results on four instruction-following benchmarks are presented in Table 1. Our analysis reveals that ContextIF not only substantially elevates the performance of its base models but also establishes a new state-of-the-art among open-source models of comparable scale. Key observations are as follows:

**ContextIF Unlocks Base Model Potential.** The most prominent finding is the consistent performance leap across all primary metrics. On the LLaMA3-8B-Instruct backbone, ContextIF-8B achieves a 6.24 point absolute gain in the IFEval Average score, increasing from 77.11 to 83.35. Notably, this improvement is uniformly observed across both strict and loose evaluation settings, which demonstrates the model's enhanced capability to handle varied constraint granularities. Such a substantial uplift validates the robustness of our RL-based context synthesis in refining the model's internal instruction-processing logic.

**ContextIF Surpasses Specialized Baselines.** When compared to specialized instruction-following models of the same scale, ContextIF-8B exhibits a decisive advantage. It outperforms established baselines such as SPAR-8B and UltraIF-8B, despite their heavy reliance on extensive SFT and DPO pipelines. This performance gap is particularly evident in the multi-turn scenarios of Multi-IF, where our model maintains superior performance as the dialogue progresses. Specifically, at the final turn, ContextIF-8B surpasses the strongest same-scale baseline, suggesting that optimizing for task-optimal context provides a more stable signal for sustaining precise instruction-following performance across extended interactions than traditional preference-based alignment.

**ContextIF Bridges the Scaling Gap.** Remarkably, ContextIF-8B effectively narrows the performance disparity traditionally seen between small-scale models and their significantly larger counterparts. On the IFEval benchmark, our model performs on par with models that possess nearly an order of magnitude more parameters, such as LLaMA3-70B-Instruct. This trend is further amplified on LiveBench, where ContextIF-8B dramatically outperforms its foundation model and all samescale specialized baselines in complex scenarios. These results provide compelling evidence that high-quality, task-optimal contexts serve as a superior signal for generalizable instruction-following,

enabling smaller open-source models to emulate the sophisticated instruction-following capabilities typically reserved for proprietary giants like GPT-4o.

In summary, the results consistently validate our approach. By training a model to generate optimized context via our multi-faceted reward signal, ContextIF significantly elevates the instruction-following performance of its base models. It not only establishes leadership among models of a similar scale but also demonstrates a remarkable ability to compete with, and in terms of real-world generalization, surpass, proprietary and much larger models. This makes ContextIF a highly effective and parameter-efficient solution for advancing state-of-the-art instruction-following. We demonstrate the robustness of our approach with consistent improvements on Mistral-7B-Instruct, with detailed results provided in Section E.2.

## 4.5 COMPARISON WITH ICL STRATEGIES

Table 2: Performance comparison of ContextIF against various ICL strategies on the LLaMA3-8B-Instruct model.

| Model | IFEval | | | | | Multi-IF | | | FollowBench | LiveBench |
|---|---|---|---|---|---|---|---|---|---|---|
| | P (L) | I (L) | P (S) | I (S) | Avg. | Turn1 | Turn2 | Turn3 | SSR | Score |
| LLaMA3-8B-Instruct | 77.02 | 84.05 | 69.44 | 77.94 | 77.11 | 63.83 | 52.24 | 43.92 | 62.90 | 46.70 |
| + zeroshot | 75.42 | 82.73 | 72.58 | 80.14 | 77.72 | 68.37 | 55.82 | 46.25 | 63.12 | 48.40 |
| + select-context | 74.31 | 82.37 | 71.35 | 79.86 | 76.97 | 61.23 | 51.14 | 42.21 | 60.43 | 43.70 |
| + LLM-context | 78.87 | 85.17 | 76.52 | 83.33 | 80.97 | 70.82 | 57.51 | 49.63 | 65.25 | 50.50 |
| + tuneLLM-context | 78.27 | 84.49 | 75.97 | 82.61 | 80.34 | 69.85 | 57.26 | 49.43 | 64.37 | 50.30 |
| + GPT4o-context | 82.44 | 88.13 | 76.71 | 83.81 | 82.77 | 73.75 | 61.94 | 52.13 | 67.38 | 57.40 |
| ContextIF-8B | **83.54** | **88.72** | **77.07** | **84.05** | **83.35** | **74.32** | **62.58** | **53.51** | **69.37** | **59.90** |

To validate the effectiveness of our proposed ContextIF framework, we conducted a comprehensive comparison of various context generation strategies, with results presented in Table 2. We first observe that while a well-prompted zero-shot baseline can slightly improve the instruction-following capabilities of the LLaMA3-8B-Instruct model, employing randomly selected in-context demonstrations consistently degrades performance across all benchmarks. Our analysis suggests that randomly chosen contexts often introduce significant distracting information, negatively impacting the model's ability to follow instructions and underscoring that effective context is paramount. The viability of automatic high-quality context generation is established when using the LLM itself to generate context, an effect that is further amplified when employing a much larger model like GPT-4o. However, when we fine-tuned a dedicated LLaMA3-8B-Instruct model on 400 curated examples to specialize in context generation, its performance paradoxically underperformed the base LLM context generator. A closer analysis reveals overfitting on specific constraint types and diminished demonstration quality, highlighting the inherent limitations of a purely supervised approach for complex context synthesis.

In contrast, ContextIF-8B establishes a new state-of-the-art across all evaluated benchmarks, surpassing all alternative ICL strategies including those powered by the significantly larger GPT-4o. It achieves superior performance across all IFEval metrics and exhibits remarkable stability on the challenging multi-turn Multi-IF benchmark, where it maintains the highest accuracy at the final turn. Furthermore, its leading performance on LiveBench and FollowBench provides compelling evidence that our RL approach transcends mere context generation by discovering a task-optimal context strategy. The fact that this learned policy proves superior even to larger, general-purpose models suggests that specialized context optimization via RL is a more effective paradigm, opening up a new and more efficient frontier for enhancing instruction-following capabilities.

## 4.6 ANALYSIS ON UNSEEN CONSTRAINTS

To evaluate the generalization capability of ContextIF, we investigate its performance on constraint types that were either absent or intentionally excluded during the training phase. While our training data primarily focuses on content, style, and format, we specifically withheld "Length" and "Keywords" constraints from the training sets of both ContextIF and the SPAR-SFT-DPO baseline to

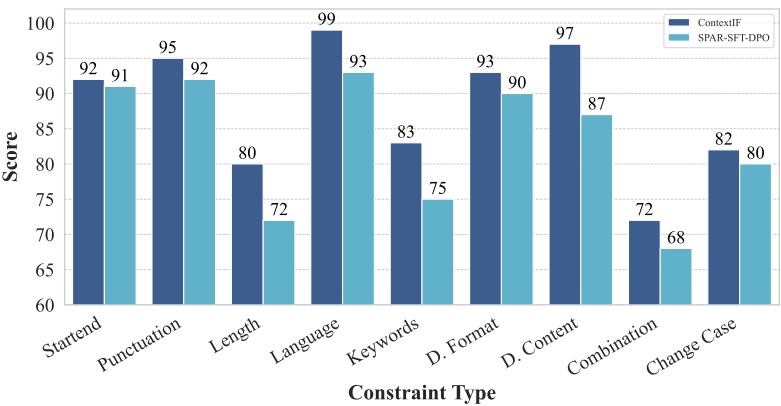

Figure 3: Prompt-level strict scores across different types of constraints on IFEval.

create a rigorous zero-shot testing environment. Following this setup, we created specialized test subsets from the IFEval benchmark focusing specifically on these unseen categories alongside other rare constraint types.

As illustrated in Figure 3, ContextIF demonstrates robust generalization, consistently outperforming the baseline across all unseen categories. Notably, on the "Language" constraint, our model achieves a near-perfect score, while on "Keywords" and "Length", it maintains a decisive 8-point lead over the baseline. These results are particularly significant because neither model received explicit supervision for these constraints. This suggests that the RL process in ContextIF does more than pattern matching; by iteratively deconstructing and reconstructing instruction-context pairs, the model acquires a fundamental and transferable understanding of instruction-following that generalizes to novel constraint spaces.

This strong zero-shot performance is complemented by significant gains in categories aligned with our training distribution, such as "D. Content" and "D. Format." In particular, on content-based tasks, ContextIF surpasses the baseline by a margin of 10 points. This dual success, combining broad generalization with specialized precision, indicates that our framework fosters a resilient instruction-following mechanism. These findings suggest that while incorporating a more diverse range of constraint types into the training data could further enhance performance, the ContextIF framework itself already cultivates a robust and generalizable instruction-following capability. We encourage the research community to further explore how dynamically generated context can serve as a powerful tool for advancing generalization in LLMs.

## 4.7 ANALYSIS ON GENERAL CAPABILITIES

A crucial aspect of our investigation is to determine whether enhancing instruction-following capabilities via ContextIF compromises the model's foundational general abilities, a phenomenon often referred to as the "alignment tax" in traditional SFT. To verify this, we evaluate our models across four diverse and challenging domains: general knowledge (MMLU), reasoning (BBH), mathematical problem solving (GSM8K), and coding (HumanEval).

Table 3 shows the performance of ContextIF in comparison to the original LLaMA3-8B-Instruct and the SFT baseline (SPAR-SFT). As hypothesized, the SPAR-SFT baseline exhibits consistent performance degradation across all benchmarks, underscoring the risk of catastrophic forgetting associated with standard fine-tuning; the most significant drop is observed in GSM8K, with a decline of 1.0 point. In stark contrast, ContextIF not only preserves but actually enhances performance across every evaluated domain. Notably, it achieves its most significant improvements in knowledge and reasoning, with a 1.7-point increase on MMLU and a 1.6-point gain on BBH over the base LLM.

These results strongly suggest that the underlying process of our RL framework, learning to deconstruct instructions, summarize constraints, and generate logically consistent examples, does not merely teach a narrow skill. Instead, it appears to refine the model's meta-learning and reasoning pathways. We believe that empowering a model to generate its own context for learning acts as a

Table 3: Performance comparison on general capability benchmarks. We report 5-shot accuracy on MMLU, 3-shot accuracy on BBH, and Pass@1 on GSM8K and HumanEval. The numbers in parentheses indicate the performance change relative to the base model.

| Method | Knowledge | Reasoning | Math | Coding | Average |
|---|---|---|---|---|---|
| | MMLU | BBH | GSM8K | HumanEval | [All] |
| LLaMA3-8B-Instruct | 68.40 | 66.50 | 79.60 | 62.20 | 69.18 |
| SPAR-SFT | 67.80 (-0.6) | 66.10 (-0.4) | 78.60 (-1.0) | 62.00 (-0.2) | 68.63 (-0.55) |
| ContextIF-8B | 70.10 (+1.7) | 68.10 (+1.6) | 80.40 (+0.8) | 63.30 (+1.1) | 70.48 (+1.3) |

form of self-distillation, which reinforces its core capabilities, facilitating the development of a more general and versatile agent. To assess the necessity of each component within our reward design that enables this synergy, we conduct a detailed ablation study in Section E.1.

## 5 Conclusion

In this study, we introduce ContextIF, a novel RL framework that enhances the instruction-following capabilities of LLMs by dynamically generating high-quality context via Context Reward. We reveal that, unlike traditional SFT and ICL methods that depend on complex and labor-intensive data curation, guiding an LLM to generate key constraints and high-quality parallel demonstrations as context through multi-faceted reward mechanism leads to significant improvements on instruction-following tasks. Critically, extensive experiments demonstrate that ContextIF not only excels in generalizing to unseen constraints but also bypasses the risk of catastrophic forgetting typically associated with SFT, preserving and even enhancing the model's general capabilities. This work showcases the immense potential of combining ICL with RL for instruction-following, and we believe empowering models to craft their own context represents a scalable and effective path toward more generalizable and aligned LLM agents.

## 6 Acknowledgments

This work is supported by the National Natural Science Foundation of China (72204087), the Shanghai Planning Office of Philosophy and Social Science Youth Project (2022ETQ001), the Chenguang Program of Shanghai Education Development Foundation and Shanghai Municipal Education Commission (23CGA28), the Shanghai Pujiang Program (23PJC030), Young Elite Scientists Sponsorship Program by CAST (YESS20240562), and the Fundamental Research Funds for the Central Universities, China. We also appreciate the constructive comments from the anonymous reviewers.

### Ethical Statement

The research conducted in this paper adheres to the ICLR Code of Ethics. Our work primarily focuses on enhancing the instruction-following capabilities of existing open-source LLMs. The datasets used for training and evaluation in our ContextIF framework were derived from publicly available instruction-following benchmarks, which consist of general-purpose, non-personal queries. We have made efforts to filter any potentially harmful or personally identifiable information during our data processing stage.

### Reproducibility Statement

We are committed to ensuring the reproducibility of our research. All datasets used for training and evaluation in this work, including IFEval, Multi-IF, FollowBench, and LiveBench, are publicly available benchmarks. The implementation details of our ContextIF framework, including the reward functions and the GRPO training algorithm, are described in Section *Methodology*. Our ContextIF framework is implemented on top of the publicly available veRL reinforcement learning library. The base models used in our experiments, including LLaMA3-8B-Instruct, are open-source and can be accessed via the Hugging Face Hub.

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

## A    USE OF LLMS

In the preparation of this manuscript, LLMs were utilized as a general-purpose writing assistance tool. Specifically, we employed LLM-based services to aid in proofreading, grammar correction, and rephrasing of sentences to improve clarity and style. The core scientific contributions, including the research ideation, experimental design, data analysis, and the formulation of conclusions, were conceived and executed entirely by the human authors. The LLMs' role was strictly limited to that of a language polishing tool. All content, including any text modified with the assistance of an LLM, was carefully reviewed, edited, and validated by the authors, who take full responsibility for the final version of this paper.

## B    DESCRIPTION OF BASELINES

**Conifer** (Sun et al., 2024) introduces an instruction-following dataset synthesized through a three-stage pipeline. Starting with seed instructions from ShareGPT, it sequentially applies query reframing, constraint generation, and recombination to construct the final training samples. We generate responses for the public instructions using LLaMA3-70B-Instruct.

**AutoIF** (Dong et al., 2024) employs a methodology for evaluating model responses using custom Python functions. These functions automatically verify adherence to a wide variety of manually designed constraints. For this baseline, we replicate their methodology on our base model to obtain the performance results.

**UltraIF** (An et al., 2025) is a framework for creating large-scale instruction-following data. It trains a specialized model, the UltraComposer, on a corpus of existing documents and datasets. This composer model is then used to synthesize the final instruction data. We reproduce their data generation and training pipeline to establish the baseline performance for this method.

**SPAR** (Cheng et al., 2024) is a self-play framework designed to generate high-quality preference data for instruction-following tasks. It addresses the issue of irrelevant content variations in standard preference pairs by using a tree-search self-refinement process. In this approach, an LLM iteratively refines its own responses to minimize distractions, creating cleaner and more comparable preference pairs for training. For our baseline, we fine-tune our base model using their officially released SFT and DPO datasets.

## C    EVALUATION BENCHMARKS

**IFEval** (Zhou et al., 2023) assesses the instruction-following capabilities of LLMs through objective, machine-verifiable prompts. It consists of approximately 500 prompts covering 25 distinct types of constraints, such as formatting and keyword requirements. We report performance using both loose and strict accuracy at the prompt and instruction levels, adhering to the benchmark's standard protocol.

**Multi-IF** (He et al., 2024) extends the evaluation of instruction-following to complex multi-turn and multilingual scenarios. Built upon IFEval, it contains 4,501 conversational dialogues across various languages, each structured into three turns. For our experiments, we report the accuracy across all three conversational turns to evaluate sustained alignment.

**FollowBench** (Jiang et al., 2024) is designed to evaluate adherence to multi-level, fine-grained constraints organized into five categories: Content, Situation, Style, Format, and Example. It emphasizes a compositional design with incremental difficulty levels. We employ GPT-4-0125-preview as an automated judge to assess whether model outputs satisfy these complex, multi-layered constraints.

**LiveBench** (White et al., 2024) is a comprehensive, contamination-free benchmark featuring diverse tasks with objective ground-truth values to prevent data leakage. We utilize its dedicated instruction-following subset to provide a robust assessment of our model's capabilities on fresh, non-stale data.

**GSM8K** (Cobbe et al., 2021) is a benchmark composed of 8,500 high-quality grade school math word problems. It is specifically designed to challenge the multi-step mathematical reasoning ca-

pabilities of language models. For our evaluation, we report the model's overall accuracy on this dataset.

**HumanEval** (Chen et al., 2021) is a code generation benchmark consisting of 164 handcrafted programming problems. Each problem includes a function signature, a docstring, and a set of unit tests (averaging 7.7 per problem) for evaluation. The benchmark assesses a model's ability to synthesize functional code from natural language descriptions, thereby testing a combination of language comprehension, reasoning, and algorithmic skills. We report the Pass@1 metric in our experiments.

**BBH** (Suzgun et al., 2023) is a challenging subset of the BIG-Bench benchmark, curated to include 23 tasks that current language models find difficult. The benchmark contains a total of 6,511 examples and is designed to rigorously evaluate a model's multi-step reasoning and problem-solving abilities. We report the accuracy metrics in our experiments.

**MMLU** (Hendrycks et al., 2021) serves as a key benchmark for assessing the breadth of a language model's world knowledge and problem-solving abilities. The benchmark is composed of multiple-choice questions that span 57 distinct tasks. These tasks cover a wide spectrum of subjects, ranging from humanities like U.S. history to professional fields such as law and medicine. In the experiments, we report the average accuracy across all tasks.

## D EXPERIMENTAL DETAILS

### D.1 PROMPTS OF CONTEXTIF

For the Actor Model, we use the following template during inference to incorporate either a zero-shot or a one-shot context:

---

**Prompt Template for Response Generation Zero-shot**

You are an expert tasked with answering the given query. Please provide a clear and concise response directly, without introductory phrases such as "What a great question", "Here is the answer", or similar expressions.
Focus solely on addressing the query. Now please answer the given query while strictly following its inside constraints.
**Query:** {query}

---

**Prompt Template for Response Generation One-shot**

You are an expert tasked with answering the given query. Please provide a clear and concise response directly, without introductory phrases such as "What a great question", "Here is the answer", or similar expressions.
Focus solely on addressing the query. Now please answer the given query while strictly following its inside constraints.

Here is an example of the constraints contained therein and one that contains constraints of the same type.

**Constraints:** {constraint}
**Example:** {Example}

Here is the Query:
**Query:** {query}

---

For the Policy Model, we use the following prompt template to generate constraint summaries and corresponding demonstrations for given queries:

---

**Prompt Template for Context Generation**

You are an expert in identifying constraints in instructions and constructing high-quality, parallel examples. Your objective is to analyze a given user query, identify its constraints, and then generate a new, similar example. Your final output MUST be a single, valid XML block and nothing else.

The required XML format is: <constraint>Summary of constraints</constraint>
<question>Newly generated question with similar constraints</question>
<answer>Answer that follows the new question's constraints</answer>

Here are the examples of correct execution you must follow:

**Example 1:**
**Input:** Generate a five-sentence story using the following words, need, money, computer, bought, happy.
**Output:**
<constraint>Generate a five-sentence story using the specified words</constraint>
<question>Generate a five-sentence story using the following words, cat, garden, sunshine, play, joy</question>
<answer>The cat wandered into the garden on a sunny day. It began to play among the flowers with great joy. The warm sunshine made the garden feel magical. Birds chirped happily as the cat chased butterflies. At the end of the day, the cat rested peacefully, content and joyful.</answer>

**Example 2:**
**Input:** Write an extravagant session plan to learn about java. Make sure to include a postscript starting with P.P.S.
**Output:**
<constraint>Include a postscript starting with P.P.S in the session plan</constraint>
<question>Write a detailed study plan to learn Python. Make sure to include a postscript starting with P.P.S.</question>
<answer>Study Plan to Learn Python:\n\n1. Introduction to Python: Understand the basics of Python syntax and data types.\n2. Control Structures: Learn about loops, conditionals, and functions.\n3. Data Structures: Explore lists, dictionaries, sets, and tuples.\n4. Object-Oriented Programming: Understand classes, objects, and inheritance.\n5. Modules and Packages: Learn how to organize code and use libraries.\n6. Practical Projects: Build small projects like a calculator or a to-do app to apply knowledge.\n\nP.P.S. Remember to practice coding daily and explore online resources to deepen your understanding.</answer>

**Example 3:**
**Input:** List the pros and cons of using two different names for the same thing. Make sure the word synonyms appears at least 3 times.
**Output:**
<constraint>The word "synonyms" must appear at least 3 times in the response</constraint>
<question>List the advantages and disadvantages of using multiple terms for the same concept. Make sure the word synonyms appears at least 3 times.</question>
<answer>Using multiple terms as synonyms for the same concept can have both pros and cons. On the positive side, synonyms can enrich communication by providing variety and preventing repetition, making the text more engaging. Additionally, synonyms can help clarify meaning by offering alternative expressions that might resonate better with different audiences. However, the use of synonyms can also lead to confusion if the terms are not perfectly interchangeable or if the audience is unfamiliar with some of the synonyms used. Moreover, excessive reliance on synonyms might dilute the precision of the message. Overall, while synonyms enhance language flexibility, careful consideration is needed to maintain clarity.</answer>

---

**Prompt Template for Context Generation**

**Example 4:**
**Input:** Answer must be in lowercase letters only throughout the entire response.
**Output:**
<constraint>Answer must be in lowercase letters only throughout the entire response</constraint>
<question>What is another word for paintings? Answer in lowercase letters only, throughout your entire answer.</question>
<answer>artworks</answer>

---

You will now be given a user query. Your task is to process this query according to the rules and examples above.
**CRITICAL INSTRUCTION:** Do NOT use generic placeholder text like "Constraint description goes here", "Question description goes here", or any similar variations in your output. You must generate specific, new content derived from the user query.
**Query:** {query}
Now, based on the user query provided above, generate your response in the specified XML format.

---

## D.2 IMPLEMENTATION DETAILS

For our primary experiments, both the policy model and the actor model were initialized from the same base model. The policy model was then fine-tuned using the GRPO algorithm as described in Section 3.4. The actor model's weights were kept frozen during all stages of training and evaluation to serve as the target LLM for instruction-following. This decoupled setup ensures that our method enhances instruction-following capabilities without degrading the model's pre-existing general knowledge.

# E ADDITIONAL EXPERIMENT RESULTS

## E.1 ABLATION STUDIES

We conduct ablation studies to investigate how each component of our context reward contributes to the model's instruction-following capabilities and to assess their necessity. The results are presented in Table 4. In the "w/o Format" setting, the format reward is removed. In the "w/o Summary", "w/o Demoq", and "w/o Demoa" settings, the rewards for the constraint summary, demonstration question, and demonstration answer are respectively excluded. As shown in Table 4, the demonstration answer reward (w/o Demoa) and the constraint summary reward (w/o Summary) prove to be the most critical components for enhancing complex instruction-following. Removing the answer faithfulness reward results in the most severe performance drop across all benchmarks, with the IFEval average score plummeting from 83.35 to 78.50. The removal of the summary reward similarly leads to significant performance degradation.

Perhaps the most insightful finding is the relative impact of the Format Reward. While its omission degrades performance, the drop is notably less severe than when the summary or faithfulness signals are removed. This indicates that while a consistent XML structure provides a beneficial scaffolding, the core challenge of instruction-following lies in the semantic interpretation and execution of intent. Our base model, already proficient in structured outputs, derives the greatest benefit from rewards that target this deeper cognitive alignment.

Ultimately, these results validate our design philosophy: the Constraint Summary Reward trains the model to accurately deconstruct an instruction, while the Answer Faithfulness and Question Relevance rewards guide the faithful reconstruction of those constraints into coherent demonstrations. It is this learned synergy of deconstruction and reconstruction, rather than mere structural mimicry, that proves essential for true instruction-following. This validates our multi-faceted reward mechanism as a necessary and unified system for enhancing model capabilities.

Table 4: Ablation results for the different components of our context reward.

| Model | IFEval Avg. | Multi-IF Turn3 | FollowBench SSR | LiveBench Score |
|---|---|---|---|---|
| ContextIF-8B | 83.35 | 53.51 | 69.37 | 59.90 |
| *w/o* Format | 81.13 | 51.21 | 67.48 | 56.10 |
| *w/o* Summary | 79.22 | 50.88 | 65.05 | 52.10 |
| *w/o* Demoq | 81.15 | 50.93 | 66.11 | 55.80 |
| *w/o* Demoa | 78.50 | 49.15 | 64.20 | 51.70 |

## E.2 CROSS-ARCHITECTURE ROBUSTNESS.

To further validate the model-agnostic nature of the ContextIF framework, we replicated our training and evaluation pipeline on the Mistral-7B-Instruct backbone. The results, summarized in Table 5, demonstrate that the superiority of our approach is not architecture-dependent. ContextIF-7B consistently and significantly outperforms all existing 7B-scale models across every evaluated benchmark, effectively establishing a new state-of-the-art for this scale.

The performance gains are particularly pronounced in fine-grained and multi-turn scenarios. On the IFEval strict instruction-level accuracy and the challenging third turn of Multi-IF, ContextIF-7B achieves substantial improvements over established specialized models like SPAR-7B and UltraIF-7B. Furthermore, it maintains a leading edge on comprehensive benchmarks like LiveBench, confirming its robust generalizability. These findings strongly indicate that the core mechanism of ContextIF, optimizing for task-specific context via a multi-faceted reward signal, is a fundamental and transferable technique. It successfully unlocks latent instruction-following capabilities regardless of the underlying model's pre-training or alignment history, underscoring its potential as a general-purpose paradigm for LLM enhancement.

Table 5: Evaluation results of different models on IFEval, Multi-IF, FollowBench (SSR), and LiveBench datasets. **P** and **I** stand for **P**rompt and **I**nstruction levels, respectively. **S** and **L** represent **S**trict and **L**oose metrics for IFEval. For LiveBench, we only report the performance on the subset of instruction-following data.

| Model | IFEval | | | | Multi-IF | | | FollowBench | LiveBench |
|---|---|---|---|---|---|---|---|---|---|
| | P (L) | I (L) | P (S) | I (S) | Turn1 | Turn2 | Turn3 | SSR | Score |
| *Mistral-7B-Instruct Models* | | | | | | | | | |
| Mistral-7B-Instruct | 53.30 | 64.19 | 48.49 | 59.31 | 46.16 | 34.53 | 28.62 | 60.87 | 50.20 |
| Conifer-7B | 54.89 | 64.98 | 50.46 | 60.91 | 46.20 | 46.27 | 34.74 | 61.57 | 51.80 |
| UltraIF-7B | 54.93 | 65.22 | 52.86 | 62.23 | 48.46 | 44.25 | 42.58 | 61.41 | 50.50 |
| AutoIF-7B | 54.90 | 65.23 | 52.87 | 62.23 | 47.63 | 37.41 | 32.58 | 59.50 | 51.60 |
| SPAR-7B | 58.25 | 68.11 | 56.56 | 66.19 | 51.42 | 48.46 | 38.13 | 67.13 | 53.70 |
| ContextIF-7B | **64.51** | **73.26** | **61.30** | **70.18** | **59.34** | **51.26** | **43.43** | **68.87** | **54.70** |

## E.3 ROBUSTNESS OF THE JUDGE MODEL

A potential concern in reward-driven learning is whether the policy model merely overfits the idiosyncratic preferences or stylistic biases of a specific judge model. To verify the objectivity and robustness of ContextIF, we conduct a systematic analysis by replacing our primary verifier, LLaMA3-70B-Instruct, with a judge from a completely different model family.

Specifically, we replaced the original LLaMA3-based judge with Qwen-2.5-72B-Instruct, a model with a distinct architecture, parameter scale, and alignment methodology. As shown in Table 6, the results are remarkably consistent across both configurations. The policy trained with Qwen-based rewards achieved an IFEval Average score of 83.38, mirroring the 83.35 achieved by the LLaMA-based version. Both configurations significantly outperform the base LLaMA3-8B-Instruct model. These findings demonstrate that ContextIF does not overfit to a specific judge's stylistic preferences. Instead, the framework leverages the judge solely for objective constraint verification—a task-agnostic capability that remains stable across high-performing LLMs. Furthermore, since our

final benchmark evaluations (*e.g.*, IFEval) rely on independent, rule-based or external verifiers, the risk of circularity is effectively eliminated. This confirms that ContextIF learns a robust and transferable heuristic for instruction-following that transcends the choice of a specific reward model.

Table 6: Performance consistency across different judge models on the IFEval benchmark.

| Model | P (L) | I (L) | P (S) | I (S) | Avg. |
|---|---|---|---|---|---|
| LLaMA3-8B-Instruct | 77.02 | 84.05 | 69.44 | 77.94 | 77.11 |
| ContextIF-8B (LLaMA3-70B Judge) | 83.54 | 88.72 | 77.07 | 84.05 | 83.35 |
| ContextIF-8B (Qwen-2.5-72B Judge) | 83.58 | 88.74 | 77.11 | 84.12 | 83.38 |

### E.4 EFFICIENCY AND METHODOLOGICAL ADVANTAGES

A critical consideration in enhancing instruction-following is the cost-benefit trade-off between generating auxiliary context and direct parameter tuning (*e.g.*, SFT or direct RLHF). To evaluate the practical utility of ContextIF, we analyze its training efficiency, inference overhead, and performance relative to direct optimization strategies.

First, we examine **training efficiency**. Unlike prior baselines such as SPAR or AutoIF, which rely on massive synthesized datasets ($\sim$50k+ samples) and complex iterative SFT/DPO pipelines, ContextIF is remarkably resource-efficient. Our RL phase utilizes only **4,000 unlabeled queries**. As summarized in Table 7, ContextIF avoids the heavy data dependency and iterative parameter updates that often lead to catastrophic forgetting. Furthermore, while generating XML-formatted context introduces a fixed token overhead during inference, it remains significantly more scalable than iterative "judge-and-refine" methods that necessitate multiple full-model rollouts and runtime evaluations for every query.

Table 7: Efficiency comparison between ContextIF and leading baselines.

| Metric | SPAR (Iterative SFT+DPO) | ContextIF (Ours) |
|---|---|---|
| Training Data | $\sim$50k+ samples (Synthesized) | **4k unlabeled queries** |
| Training Complexity | Multi-stage SFT + Iterative DPO | **One-time RL (GRPO)** |
| Inference Latency | High (Multi-turn Refinement) | **Fixed (Single Forward Pass)** |

To further justify the necessity of learning a context strategy over direct parameter fitting, we conducted a **compute-matched experiment**. We trained the LLaMA3-8B-Instruct model via GRPO to answer instructions **directly** (*Direct-RL*), utilizing the identical 4,000 samples and the same LLaMA3-70B-Instruct reward model. As illustrated in Table 8, while Direct-RL yields improvements in instruction-following, it significantly lags behind ContextIF.

Table 8: Comparison between ContextIF and a compute-matched Direct-RL baseline. Both models are trained using the same 4k queries and identical reward signals.

| Model | IFEval (Avg) | MMLU | BBH | GSM8K | HumanEval |
|---|---|---|---|---|---|
| LLaMA3-8B-Instruct | 77.11 | 68.40 | 66.50 | 79.60 | 62.20 |
| Direct-RL (GRPO) | 80.74 | 68.20 | 66.20 | 79.50 | 62.10 |
| **ContextIF-8B (Ours)** | **83.35** | **70.10** | **68.10** | **80.40** | **63.30** |

Crucially, Direct-RL exhibits a noticeable "alignment tax," resulting in slight performance regressions across general reasoning benchmarks (MMLU and BBH). In sharp contrast, ContextIF preserves and even enhances foundational capabilities. These results confirm that optimizing a policy to generate its own learning context is a far more robust and effective paradigm than direct task-specific tuning, effectively bypassing the limitations of standard alignment techniques.

