# OpenReview forum: "ContextIF: Enhancing Instruction-Following through Context Reward"
_ICLR.cc/2026/Conference — ICLR 2026 Poster_

### Official Review · Reviewer_KBRG · 2025-10-16

**Soundness:** 2
**Presentation:** 2
**Contribution:** 2
**Rating:** 4
**Confidence:** 3

**Summary:**

This paper presents **ContextIF**, a reinforcement learning framework that enables LLMs to **dynamically generate high-quality, task-specific contexts** for instruction following.
Unlike static SFT or ICL methods, ContextIF trains a **context generator** guided by a **multi-dimensional Context Reward** and optimized with **GRPO**.
Experiments show that an **8B model** trained with ContextIF can **match or surpass** much larger models (e.g., 70B, GPT-4o) across benchmarks.

**Strengths:**

- **Novel and effective framework:**
  Integrates RL with ICL for dynamic context generation, effectively addressing the static nature of traditional fine-tuning or ICL.
- **Comprehensive experiments:**
  Strong performance and generalization results with clear analysis on catastrophic forgetting and parameter efficiency.

**Weaknesses:**

1. **Motivation:**
   The *Introduction* mentions “existing studies” on automatic context generation but lacks citations and fails to clarify how **ContextIF** differs from prior methods.

2. **Contribution clarity:**
   The main novelty lies in the **Constraint Reward**, but the paper lacks theoretical justification for the `<constraint>`, `<question>`, `<answer>` structure and the reward components r_summary, r_demoq, r_demoa.

3. **Experimental issue:**
   In **Table 1**, **P(S)** and **I(S)** are highlighted even though they underperform **SPAR-8B**, which seems inconsistent.

4. **Other concerns:**
   - The generator’s performance ceiling depends on the **judge model**, possibly transferring its biases.
   - Using a **70B model** as a reward provider for training an **8B model** is **computationally expensive** and limits scalability.

**Questions:**

See Weakness.

---

> ### Author Response · Authors · 2025-12-02
> **Response to Reviewer KBRG**
>
> Dear Reviewer KBRG,
>
> Thank you for your thoughtful and constructive feedback. We deeply appreciate your suggestions regarding the motivation clarity, reward design justification, and experimental consistency. We hope our detailed response could address your concerns:
>
> ### 1. Concern about Motivation (W1)
>
> > **Weaknesses 1:** The Introduction mentions “existing studies” on automatic context generation but lacks citations and fails to clarify how ContextIF differs from prior methods.
>
> Thank you for pointing out the shortcomings in our writing and referencing. We have updated the Introduction to cite relevant works and clarified the distinctions.
> Prior research primarily focuses on retrieval or naive self-generation. However, retrieval fails for unseen, complex, multi-level instructions where matching contexts do not exist. Furthermore, unoptimized self-generation without quality verification is often unreliable, lacking qualified structural and constraint semantic information, and can even degrade performance (Table 2). ContextIF addresses this by employing RL with Verifiable Rewards, ensuring the dynamic synthesis of high-quality, instruction-following demonstrations.
>
> ### 2. Concern about Contribution Clarity (W2)
>
> > **Weaknesses 2:** The main novelty lies in the Constraint Reward, but the paper lacks theoretical justification for the `<constraint>`, `<question>`, `<answer>` structure and the reward components r_summary, r_demoq, r_demoa.
>
> Thanks for your valuable feedback regarding the theoretical foundations of our method. Our structural design is empirically grounded in ICL experiments. To provide direct evidence, we present a structural ablation study of IFEval below. We found that adding either an LLM-generated constraint summary or a demonstration to the LLaMA3-8B-Instruct base model improves instruction-following, but combining them into a full context yields the most significant and synergistic gains.
>
> | IFEval                      | P(L)  | I(L)  | P(S)  | I(S)  | Avg   |
> | :-------------------------- | :---- | :---- | :---- | :---- | :---- |
> | LLaMA3-8B-Instruct          | 77.02 | 84.05 | 69.44 | 77.94 | 77.11 |
> | + zeroshot                  | 75.42 | 82.73 | 72.58 | 80.14 | 77.72 |
> | + zeroshot-LLMconstraint    | 76.24 | 82.97 | 74.26 | 81.39 | 78.72 |
> | + zeroshot-LLMdemonstration | 76.81 | 83.37 | 75.23 | 82.15 | 79.39 |
> | + zeroshot-LLMcontext       | 78.87 | 85.17 | 76.52 | 83.33 | 80.97 |
> | SPAR-8B-SFT-DPO-iter1       | 78.01 | 84.67 | 75.82 | 82.60 | 80.27 |
>
> Notably, this combined context, generated without any training, is so effective that it surpasses some fully trained SFT-DPO methods, highlighting the immense potential of this structure for instruction-following tasks.
> Moreover, This finding is strongly corroborated by our original reward ablation in Table 4, which shows that the rewards are indeed critical components for unlocking the context's full potential.
>
> ### 3. Concern about Clarification of Writing Issue (W3)
>
> > **Weaknesses 3:** In Table 1, P(S) and I(S) are highlighted even though they underperform SPAR-8B, which seems inconsistent.
>
> Thank you for your careful and detailed reading. This was a presentation mistake on our part. We have corrected this issue and thoroughly reviewed the entire paper to address any other errors.

---

> > ### Author Response · Authors · 2025-12-02
> >
> > ### 4. Concern about Clarification of Experimental Details (W4)
> >
> > > **Weaknesses 4:** The generator’s performance ceiling depends on the judge model, possibly transferring its biases.
> >
> > Thank you for your insightful comment. Unlike traditional distillation that mimics teacher outputs, our framework is built on the principle that judging is a much simpler and more reliable task for an LLM than open-ended generation.
> > To empirically verify this robustness, we conducted an ablation study replacing the LLaMA3-70B-Instruct judge with Qwen-2.5-72B-Instruct, a model from a completely different family with distinct architectures and alignment methodologies. As shown in the table below, the performance is remarkably consistent: the Policy trained with the Qwen-based reward achieved an IFEval average of 83.38, virtually identical to the Llama-based version 83.35. This consistency confirms that ContextIF does not overfit to specific judge biases but instead learns a robust, model-agnostic heuristic for instruction-following.
> >
> > | IFEval                        | P(L)  | I(L)  | P(S)  | I(S)  | Avg   |
> > | :---------------------------- | :---- | :---- | :---- | :---- | :---- |
> > | LLaMA3-8B-Instruct            | 77.02 | 84.05 | 69.44 | 77.94 | 77.11 |
> > | ContextIF-8B-LLaMA3-70B-Ins   | 83.54 | 88.72 | 77.07 | 84.05 | 83.35 |
> > | ContextIF-8B-Qwen-2.5-72B-Ins | 83.58 | 88.74 | 77.11 | 84.12 | 83.38 |
> >
> > > **Weaknesses 5:** Using a 70B model as a reward provider for training an 8B model is computationally expensive and limits scalability.
> >
> > Thanks for your critical remarks on the computational cost and scalability of our approach. To achieve our goal of enhancing instruction-following, we posit that utilizing a 70B model to generate reward signals is substantially more cost-effective and scalable than either manual annotation or the massive data synthesis required for comprehensive SFT and preference learning.
> > Furthermore, most instruction-following methods typically leverage powerful models for supervision; both UltraIF (LLaMA3.1-70B-Instruct) and AutoIF (LLaMA3-70B-Instruct) use 70B-scale models to generate their massive training data. SPAR requires separately trained judge and refinement model, plus an additional revision step at inference time.
> > In contrast, our method demonstrates remarkable efficiency. We achieve superior results with a training set of only 4,000 queries by tasking the 70B model with the comparatively simpler role of judging rather than data generation, which allows our 8B model to outperform these heavily trained baselines and underscores the practical scalability of our framework.

---

### Official Review · Reviewer_EDdA · 2025-10-16

**Soundness:** 3
**Presentation:** 2
**Contribution:** 3
**Rating:** 6
**Confidence:** 4

**Summary:**

This work proposes **ContextIF**, a reinforcement learning framework to automate and optimize context generation for in-context learning (ICL) in instruction-following tasks. The core idea is to train a policy model to act as a context generator that, given a user query, produces a self-contained ICL demonstration consisting of: (1) a precise constraint summary and (2) a corresponding question-answer pair. The policy is optimized using Group Relative Policy Optimization (GRPO) guided by a composite context reward that evaluates both structural correctness (format reward) and semantic quality (constraint reward).

The experiment framework is based on several instruction-following benchmarks. And the proposed method ranks the top.

**Strengths:**

1. The proposed method can have strong generalization to unseen constraint types (e.g., Language, Keywords, Length), which is a critical advantage over supervised methods.
2. ContextIF maintains and even enhances the base model's performance on general benchmarks, because it does not change the parameters of the Actor Model.

**Weaknesses:**

1. **Lack of clarity on model and training details:** Which model serves as the Policy Model? The paper does not explicitly state which model is used as the policy (e.g., is it initialized from Llama-3-8B-Instruct, a separate smaller model, or the same base model?).

2. **Decoupling of Policy and Actor models is not well justified:** The paper trains the Policy Model to generate context but keeps the Actor Model (the target LLM) frozen. While this avoids catastrophic forgetting, it raises questions:
	- Why not directly train the Generator Model using the same context reward based on the GRPO?
	- What if the policy and actor are the same model? Would end-to-end training be more effective?

**Questions:**

same as above.

---

> ### Author Response · Authors · 2025-12-02
> **Response to Reviewer EDdA**
>
> Dear Reviewer EDdA,
>
> Thank you for your detailed and constructive feedback. We really value your recognition of ContextIF’s generalization to unseen constraints and your comments on the policy model initialization and the decoupled training strategy. We hope our detailed response could address your concerns:
>
> ### 1. Concern about Training Details (W1)
>
> > **Weakness 1:** Lack of clarity on model and training details: Which model serves as the Policy Model? The paper does not explicitly state which model is used as the policy (e.g., is it initialized from Llama-3-8B-Instruct, a separate smaller model, or the same base model?).
>
> Thanks for highlighting the missing details in our paper. As highlighted in our Introduction (Paragraph 3), we explicitly identified the critical bottleneck of naive self-generation for instruction-following. This observation serves as one of the foundational motivations for our work. For all experiments, the Policy Model is initialized from the same base model as the Actor Model. Specifically, for our main results, the Policy Model is a Llama-3-8B-Instruct model that is then fine-tuned via GRPO to specialize in the task of context generation. Similarly, for the experiments in the appendix, the Policy Model is initialized from Mistral-7B-Instruct.
> The Actor Model is a separate, frozen instance of the same base model. We have revised Section 3.1 and provided additional details in Appendix to make this crucial detail explicit and unambiguous.
>
> ### 2. Concern about the Justification for Decoupling (W2)
>
> > **Weakness 2:** Decoupling of Policy and Actor models is not well justified: The paper trains the Policy Model to generate context but keeps the Actor Model (the target LLM) frozen. This raises several questions. Why not directly train the Generator Model using the same context reward based on the GRPO?
>
> Thanks for your insightful questions regarding our methodology. Prior baselines enhance instruction-following through complex SFT and RL data construction, which entails higher costs and requirements. This process often leads to catastrophic forgetting, harms the model's general capabilities, and fails to generalize to unseen, complex, multi-level constraints, thus not meeting real-world needs.
>
> Furthermore, our experiments in Table 2 show that even without training, generating context via the model itself can achieve performance that matches or even surpasses that of models after complex SFT-DPO training. Our method encourages a higher level of meta-learning; the policy learns the abstract skill of generating effective context, which leads to superior instruction-following generalization at a lower cost.
>
> Finally, the policy model is a lightweight, "plug-and-play" module that can enhance any compatible, off-the-shelf actor model, including black-box APIs, without requiring access to their weights. This provides far greater flexibility and scalability in real-world applications.
>
> ### 3. Concern about the End-to-end training (W3)
>
> > **Weakness 3:** What if the Policy and Actor are the same model? Would end-to-end training be more effective?
>
> Thank you for your insightful comment. Our goal is to enhance instruction-following capabilities while simultaneously maintaining the model's generalizability so that it can be applied to any other task.
> The primary reason for our decoupled design is to preserve the foundational capabilities of the actor model and completely avoid catastrophic forgetting—a common pitfall of direct fine-tuning. By keeping the actor model's parameters frozen, we ensure its vast, pre-existing knowledge remains intact. Our results in Table 3 empirically validate this advantage, showing that direct fine-tuning methods degrade performance on general benchmarks, whereas our method does not. An end-to-end model, which would require updating parameters for the entire model, would be exposed to this same risk.

---

### Official Review · Reviewer_vWMh · 2025-10-28

**Soundness:** 2
**Presentation:** 2
**Contribution:** 2
**Rating:** 4
**Confidence:** 4

**Summary:**

This paper proposes ContextIF, an RL framework that generates in-context demonstrations automatically by a RL (GRPO) trained model. The reward combines a format check (structure-only) and a constraint reward scored by a judge model over three facets (summary, question parallelism, answer faithfulness). The authors compare their proposed method ContextIF with various baselines and ICL methods, which show superior performance.

**Strengths:**

- Clear improvement over baselines (ICL, base, and instruction-tuned). Across IFEval, Multi-IF, FollowBench, and LiveBench, ContextIF-8B outperforms LLaMA3-8B-Instruct and some previous SFT/DPO systems.

- Beats alternative ICL strategies, including GPT-4o-generated contexts. In the head-to-head ICL comparison, ContextIF-8B tops zero-shot, random/select-context, LLM-context, tuned-LLM-context, and GPT-4o-context.

- Proposed methodology is simple and straightforward.

**Weaknesses:**

- **Limited novelty.** The main move is to train a *policy* that emits a `<constraint>/<question>/<answer>` **XML** block and optimize it with a composite **format + constraint** reward, trained via **GRPO**. All ingredients are established yet this looks like a careful engineering bundle rather than a conceptual jump. (i.e. RLVR is known to work well, and this seems like just applying that for auto-ICL model). Could you please clarify what is *principally novel* in this work?

- **Task narrowness & scalability.** Evaluation targets **instruction-following** only (IFEval, Multi-IF, FollowBench, LiveBench *instruction-following subset*), with rewards tailored to *definitive, machine-/judge-verifiable* constraints leaving open whether this scales to harder, less judgeable tasks. One example could be *execution- or test-based** reward channels (e.g., unit tests for code, tool outcomes) or that require reasoning or planning efforts. The tasks tested here seem to be too simple compared to benchmarks used for evaluating comparatively recent open-source model performances.

- **ICL vs. training the model directly; cost/benefit unclear.** The paper motivates ICL to avoid parameter updates, yet the approach **does** train a policy with RL (GRPO) and adds **inference overhead** (generating the XML context) before answering. There is no compute/latency accounting (rollouts per query, group size, wall-clock, token overhead), so it’s hard to judge whether one should instead train the *task model* with SFT/RLHF/GRPO to answer directly, skipping auto-ICL, especially considering the **narrow coverage of the nature of the task**. Please report training/inference costs and include a **compute-matched** baseline where a model is RL-trained to *answer* (using the same judge) rather than to *generate demonstrations*.

**Questions:**

1. Please directly address the points raised in the weakness section.

2. What motivated choosing Llama-3-8B and Mistral-7B-Instruct? i.e. Can you report results on more recent open-source models and compare trends?

3. Given the narrow task scope (i.e. definite instruction following), why train an in-context generator instead of training the answerer directly under a matched compute budget? Could you include compute-controlled comparisons and token/latency overheads?

4. Why did you use GPT-4o for the demonstration comparison , instead of stronger contemporaries (e.g., GPT-4.1 or GPT-5)? If feasible, add those results or justify the choice.

5. I might have missed this in the paper, but couldn't find the details - which model did you use for reward model during training?

---

> ### Author Response · Authors · 2025-12-02
> **Response to Reviewer vWMh**
>
> Dear Reviewer vWMh,
>
> Thank you for your critical and constructive review. We particularly appreciate your challenge regarding the trade-offs between training an ICL generator versus direct model training. In response, we have conducted compute-matched comparisons and an overhead analysis to demonstrate the efficiency of our approach. We also clarify our specific contributions regarding novelty and scalability below.
>
> ### 1. Concern about Novelty (W1)
>
> > **Weaknesses 1:** Limited novelty. The main move is to train a policy that emits a `<constraint>`/`<question>`/`<answer>` XML block and optimize it with a composite format + constraint reward, trained via GRPO. All ingredients are established yet this looks like a careful engineering bundle rather than a conceptual jump. (i.e. RLVR is known to work well, and this seems like just applying that for auto-ICL model). Could you please clarify what is principally novel in this work?
>
> Thanks for your critical appraisal regarding the novelty of our approach. We respectfully contend that the principal novelty of ContextIF is not merely the combination of components, but a paradigm shift from data-centric parameter tuning to RL-driven context optimization. Traditional instruction-following methods rely on complex SFT and reinforcement learning pipelines that necessitate constructing massive, exhaustive datasets to cover diverse constraint combinations. This process is prohibitively expensive and, as shown in our experiments, prone to catastrophic forgetting (Table 3) and poor generalization to unseen multi-level constraints (Figure 3). Our work empirically reveals that an untrained base model, solely through autonomously generated context, can match or even surpass the performance of large-scale SFT-DPO models. This demonstrates that the latent potential of ICL can be unlocked to achieve superior results at a fraction of the training cost.
>
> Furthermore, unlike supervised methods that struggle to generalize to novel or complex constraints, our core innovation lies in leveraging reinforcement learning to fundamentally robustify the ICL process. By optimizing the context generation policy via a verifiable reward rather than mimicking static supervision, ContextIF avoids the overfitting inherent in SFT. This results in a robust and transferable reasoning framework capable of enhancing instruction-following under arbitrary constraints while strictly preserving the model's general capabilities (Tables 1 and 3)
>
> ### 2. Concern about Task (W2)
>
> > **Weaknesses 2:** Task narrowness & scalability. Evaluation targets instruction-following only (IFEval, Multi-IF, FollowBench, LiveBench instruction-following subset), with rewards tailored to definitive, machine-/judge-verifiable constraints leaving open whether this scales to harder, less judgeable tasks. One example could be execution- or test-based* reward channels (e.g., unit tests for code, tool outcomes) or that require reasoning or planning efforts. The tasks tested here seem to be too simple compared to benchmarks used for evaluating comparatively recent open-source model performances.
>
> Thanks for your insightful comments on the task narrowness and scalability of our approach. Instruction-following serves as the foundational capability for reliable agents. We wish to emphasize that the evaluated benchmarks (IFEval, Multi-IF, FollowBench, LiveBench) are rigorous, involving multi-turn interactions and multi-level, complex constraints. Even state-of-the-art proprietary models often fail to fully adhere to such nuanced instructions.
>
> Regarding scalability to harder reasoning tasks (e.g., coding and math), our approach demonstrates a significant advantage over direct parameter tuning. As explicitly shown in Table 3 (General Capabilities), unlike prior SFT methods that often suffer from catastrophic forgetting and degrade reasoning performance, ContextIF achieves performance gains on GSM8K (+0.8%) and HumanEval (+1.1%). This empirical evidence confirms that our method effectively scales to reasoning-intensive domains by fostering a transferable meta-skill, rather than being limited to simple constraint satisfaction.

---

> > ### Author Response · Authors · 2025-12-02
> >
> > ### 3. Concern about ICL vs. training directly (W3)
> >
> > > **Weaknesses 3:** ICL vs. training the model directly; cost/benefit unclear. The paper motivates ICL to avoid parameter updates, yet the approach does train a policy with RL (GRPO) and adds inference overhead (generating the XML context) before answering. There is no compute/latency accounting (rollouts per query, group size, wall-clock, token overhead), so it’s hard to judge whether one should instead train the task model with SFT/RLHF/GRPO to answer directly, skipping auto-ICL, especially considering the narrow coverage of the nature of the task. Please report training/inference costs and include a compute-matched baseline where a model is RL-trained to answer (using the same judge) rather than to generate demonstrations.
> >
> > Thank you for your insightful comment. We respectfully emphasize that prior instruction-following baselines rely heavily on intricate SFT and RL pipelines, where constructing a dataset that comprehensively covers all constraint types is both complex and prohibitively expensive.
> >
> > For instance, strong baselines like AutoIF and UltraIF require synthesizing tens of thousands of samples using massive models (e.g., Llama-3-70B), while methods like SPAR incur significant inference latency due to iterative judge-and-refine loops. Crucially, such continuous data distillation and iterative parameter updates render these models highly susceptible to catastrophic forgetting, significantly degrading their general capabilities (Table 3). The table below provides a detailed efficiency comparison to clarify the cost-benefit advantage of ContextIF:
> >
> > | Metric                | SPAR (Iterative SFT+DPO)     | ContextIF (Ours)          |
> > | :-------------------- | :--------------------------- | :------------------------ |
> > | **Training Data**     | ~50k+ samples (Synthesized)  | 4k queries                |
> > | **Training Cost**     | SFT + Iterative DPO          | One-time RL               |
> > | **Inference Latency** | High (Multi-turn Refinement) | Fixed (Single Model Call) |
> >
> > Moreover, our empirical results validate the inherent advantage of ICL for this task; as shown in Table 2, even the base model utilizing self-generated context without any RL training outperforms models trained with extensive SFT and DPO. Consequently, our method avoids the heavy data dependencies and alignment tax of full-parameter tuning.
> >
> > With regard to training and inference efficiency, we argue that ContextIF offers a superior cost-benefit ratio compared to direct training baselines like SPAR, excelling in both training efficiency and inference scalability. Specifically, our RL stage is highly resource-efficient, utilizing only 4,000 unlabeled queries. The training infrastructure consisted of one server equipped with 8x H20 GPUs for policy optimization and a separate server with 8x H20 GPUs for reward model deployment. This setup contrasts sharply with baselines that require massive labeled datasets and complex pipelines of SFT followed by multiple DPO iterations. Furthermore, at inference time, ContextIF incurs only a fixed, minimal latency overhead via a single Policy forward pass, avoiding the prohibitive computational costs of tree-search or iterative refinement methods—such as those employed by SPAR—that necessitate multiple rollouts and runtime judge evaluations for every single query. Thus, ContextIF achieves state-of-the-art performance and generalization without the massive data requirements or significant runtime latency associated with intensive direct training approaches.
> >
> > Regarding the compute-matched baseline, we conducted an additional experiment where we trained the LLaMA3-8B-Instruct model via GRPO to answer directly rather than generating demonstrations, utilizing the identical dataset of 4,000 samples and the same LLaMA3-70B-Instruct reward model. The comparative results are presented below:
> >
> > |                         | IFEval (Avg) | MMLU  | BBH   | GSM8K | HumanEval |
> > | :---------------------- | :----------- | :---- | :---- | :---- | :-------- |
> > | LLaMA3-8B-Instruct      | 77.11        | 68.40 | 66.50 | 79.60 | 62.20     |
> > | LLaMA3-8B-Instruct-GRPO | 80.74        | 68.20 | 66.20 | 79.50 | 62.10     |
> > | ContextIF-8B            | 83.35        | 70.10 | 68.10 | 80.40 | 63.30     |
> >
> > The results demonstrate that while direct RL training with the identical data scale and reward model yields improvements in instruction-following, it notably lags behind ContextIF. Furthermore, direct tuning incurs an alignment tax, evidenced by slight regressions across general benchmarks. In sharp contrast, ContextIF not only achieves superior instruction-following but also preserves and enhances general reasoning capabilities. This confirms that optimizing the policy to generate context is a far more effective and robust strategy than direct parameter fitting, particularly when avoiding catastrophic forgetting is a priority.

---

> > > ### Author Response · Authors · 2025-12-02
> > >
> > > ### 4. Concern about Backbone LLMs (Q2)
> > >
> > > > **Question 2:** What motivated choosing Llama-3-8B and Mistral-7B-Instruct? i.e. Can you report results on more recent open-source models and compare trends?
> > >
> > > Thank you for your valuable suggestion. We selected Llama-3-8B and Mistral-7B-Instruct primarily to facilitate a direct comparison. Previous baselines constructed their instruction-following SFT and DPO datasets using the Llama-3-series and Mistral-series. By choosing these specific series, we ensure a fair comparison where improvements are attributed to the method itself rather than architectural differences. To validate generalizability on recent architectures, we implemented ContextIF on LLaMA-3.1-8B-Instruct, with the results shown below:
> > >
> > > |                      | IFEval (Avg) | MultiIF (Turn3) | FollowBench(SSR) | LiveBench |
> > > | :------------------- | :----------- | :-------------- | :--------------- | :-------- |
> > > | LLaMA3.1-8B-Instruct | 77.68        | 51.26           | 63.41            | 57.10     |
> > > | ContextIF3.1-8B      | 83.87        | 61.17           | 69.68            | 60.70     |
> > >
> > > The results confirm that ContextIF delivers substantial gains even on the stronger, more recent LLaMA-3.1 architecture. This validates that our framework effectively mines the latent potential of ICL regardless of the underlying model version, proving its robustness and continued relevance for newer open-source LLMs.
> > >
> > > ### 5. Concern about Motivation (Q3)
> > >
> > > > **Question 3:** Given the narrow task scope (i.e. definite instruction following), why train an in-context generator instead of training the answerer directly under a matched compute budget? Could you include compute-controlled comparisons and token/latency overheads?
> > >
> > > Thank you for raising these issues. We chose the decoupled ContextIF architecture specifically to circumvent the alignment tax inherent in directly tuning the Answerer. Direct optimization (SFT/DPO) often degrades general capabilities (Table 3). Conversely, as shown in Table 2, merely generating effective self-context—without training the answerer—matches or exceeds complex SFT-DPO baselines. This indicates our method encourages high-level meta-learning, where the policy acquires the abstract skill of generating effective contexts rather than memorizing data patterns.
> > >
> > > In terms of cost, baselines often utilize larger models to design data construction pipelines covering comprehensive constraint categories. Specifically, SPAR requires three rounds of DPO iterations and relies on expensive auxiliary 'judge' and 'revise' model for inference. By contrast, ContextIF achieves superior results through a single pass of context generation, incurring significantly lower costs while demonstrating better generalization to unseen, multi-level complex constraints.
> > >
> > > ### 6. Concern about Experimental (Q4)
> > >
> > > > **Question 4:** Why did you use GPT-4o for the demonstration comparison , instead of stronger contemporaries (e.g., GPT-4.1 or GPT-5)? If feasible, add those results or justify the choice.
> > >
> > > Thank you for your valuable suggestion. We supplemented the experiments with a demonstration comparison using GPT-4.1. The results demonstrate that ContextIF-8B achieves superior performance compared to the stronger GPT-4.1 baseline across most benchmarks. While GPT-4.1 maintains a marginal lead on LiveBench, ContextIF’s overall dominance confirms that our RL-optimized policy generates more effective, task-specific contexts than even the most capable general-purpose proprietary models.
> > >
> > > |                    | IFEval (Avg) | MultiIF (Turn3) | FollowBench(SSR) | LiveBench |
> > > | :----------------- | :----------- | :-------------- | :--------------- | :-------- |
> > > | LLaMA3-8B-Instruct | 77.11        | 43.92           | 62.90            | 46.70     |
> > > | + GPT4o-context    | 82.77        | 52.13           | 67.38            | 57.40     |
> > > | + GPT4.1-context   | 83.15        | 53.27           | 69.29            | 60.10     |
> > > | ContextIF-8B       | 83.35        | 53.51           | 69.37            | 59.90     |
> > >
> > > ### 7. Concern about Reward Model (Q5)
> > >
> > > > **Question 5:** I might have missed this in the paper, but couldn't find the details - which model did you use for reward model during training?
> > >
> > > Thank you for your comment. As stated in Section 3.3, "Constraint Reward," the semantic quality is evaluated using a powerful judge model LLaMA3-70B-Instruct. To be explicit, we used LLaMA3-70B-Instruct as the judge model for all experiments. We have made this point more prominent in the revised manuscript to avoid any ambiguity.

---

### Official Review · Reviewer_EaXT · 2025-11-01

**Soundness:** 3
**Presentation:** 3
**Contribution:** 2
**Rating:** 6
**Confidence:** 3

**Summary:**

The paper introduces ContextIF, a reinforcement learning framework that trains a policy model to generate, per query, an XML-structured in-context “context block” consisting of a constraint summary and a parallel QA demonstration, optimized with a composite “context reward” combining a binary format check and a judge-model-based constraint reward under GRPO, to improve instruction-following without modifying the target LLM’s parameters.

**Strengths:**

- Clear problem framing: static retrieval or manual pool curation limits ICL for complex, compositional constraints; the proposed generator produces per-query contextual demonstrations with explicit structure and a verifiable reward, which is straightforward to implement and evaluate.​
- Method simplicity with practical stabilization: GRPO with groupwise normalized advantages, a binary format reward, and a discrete, modular constraint reward decomposed into summary, demo-question, and demo-answer checks using a strong judge model, makes the pipeline reproducible and extensible.​
- Empirical breadth: strong improvements on IFEval/Multi-IF/FollowBench/LiveBench with LLaMA3-8B and Mistral-7B, comparisons to SFT/DPO pipelines (Conifer, UltraIF, SPAR, AutoIF), and a comparison against LLM-generated and GPT-4o-generated contexts; ablations suggest specific reward components drive most gains.

**Weaknesses:**

- Reward dependence on judge models: the constraint reward relies on a large judge (e.g., Llama3-70B-Instruct), which risks overfitting to the judge’s preferences, raises circularity concerns, and complicates claims of model-agnostic gains; calibration of judge reliability and cross-judge robustness is not systematically analyzed.​
- Limited theoretical grounding: there are no guarantees that optimizing the composite reward improves downstream adherence under distribution shifts or alternative scoring; the analysis is largely empirical and does not quantify variance, bias, or brittleness in the reward signal despite discrete, thresholded scoring.​
- Baseline protocol fairness: several baselines are reproduced via the authors’ implementations; compute budgets, hyperparameter sweeps, and prompt formats may differ and could advantage ContextIF, especially vs. retrieval-based ICL with large pools or better rankers not exhaustively tuned here.​
- Scope of generalization: while unseen constraint categories are tested within IFEval-style settings, broader robustness under different prompting regimes, safety constraints, and multilingual/multi-turn mixtures beyond the chosen benchmarks is not deeply evaluated, limiting the generality claims.​
- Format coupling risk: strict XML compliance is rewarded; this may inadvertently steer models toward structural conformity rather than deeper semantic adherence, and could reduce transfer to settings without such formatting, which is not assessed with alternate structures or no-format variants.​
- Cost and scalability details: end-to-end latency and memory profiles, especially with multiple rollouts per query and judge calls, are not characterized across larger batch sizes or longer inputs; comparisons with GPT-4o-context do not clarify cost parity or throughput under matched budgets.​
Potential leakage and template bias: the generated demonstration is a “parallel” QA with similar constraints; the paper does not quantify whether distributional shortcuts or template echoing contribute to gains versus genuine constraint reasoning, nor analyze failure modes when demonstrations are adversarial or spurious.​

Safety/ethics evaluation is minimal: while datasets are public and filtering is claimed, there is no targeted analysis for harmful content amplification or reward hacking where the policy learns to game the judge without improving real instruction compliance, especially in free-form domains.​

**Questions:**

- How robust are results to replacing the judge model with different families/sizes or rule-based verifiers, and does performance persist under cross-judge evaluation to mitigate reward overfitting concerns?​
- What are the compute/latency costs per query relative to retrieval-based ICL and SFT pipelines, including rollout counts, judge inference, and GRPO updates, especially at scale for production settings?​
- Can the method drop strict XML constraints without loss, or generalize to alternative schemas (JSON, unconstrained text) while retaining gains, and how sensitive are improvements to formatting choices?​
- How does ContextIF perform under adversarial or misleading constraints, or when instruction sets intentionally include conflicting requirements; is there a failure analysis of generated context harming adherence?​
- Could combining retrieval with generation (e.g., retrieved seed plus RL-edited context) outperform pure generation, and have such hybrids been tested under matched budgets and pools?​
- For multi-turn settings, can the context generator adapt across turns with memory, and how does it interact with conversation state vs. single-turn re-generation?

---

> ### Author Response · Authors · 2025-12-02
> **Response to Reviewer EaXT**
>
> Dear Reviewer EaXT,
>
> We sincerely appreciate you taking the time and effort to evaluate our work and provide insightful feedback. Your positive ratings give us great encouragement. We provide detailed responses below to clarify your concerns.
>
> ### 1. Concern about the Robustness of the Judge Model (W1 & Q1)
>
> > **Weakness 1:** Reward dependence on judge models: the constraint reward relies on a large judge (e.g., Llama3-70B-Instruct), which risks overfitting to the judge’s preferences, raises circularity concerns, and complicates claims of model-agnostic gains; calibration of judge reliability and cross-judge robustness is not systematically analyzed.
>
> > **Question 1:** How robust are results to replacing the judge model with different families/sizes or rule-based verifiers, and does performance persist under cross-judge evaluation to mitigate reward overfitting concerns?
>
> We appreciate your attention to the robustness of the judge model and potential model biases.
>
> **From a theoretical perspective:**
>
> - ContextIF employs the judge model solely to verify the policy model's output quality, rather than requiring the policy model to imitate the judge model’s instruction-following behavior. This design is fundamentally different from distillation-based approaches.
> - Constraint verification represents a general capability of LLMs that is largely independent of their generation style and preferences.
> - Moreover, the final benchmark evaluation does not rely on any judge model, eliminating the risk of circularity in performance assessment.
>
> Therefore, using judge models from different families and sizes is not expected to introduce systematic instruction-following bias.
>
> **From an empirical perspective:**
>
> - To validate the robustness of ContextIF across different judge models, we conducted an ablation study by replacing the original LLaMA3-70B-Instruct judge with Qwen-2.5-72B-Instruct, a model from a completely different family with distinct architecture, parameter scale, and alignment methods.
> - As shown in the table below, the performance of the two judges is nearly identical: the policy trained with Qwen-based rewards achieved an average IFEval score of 83.38, compared to 83.35 for the LLaMA-based version, both surpassing LLaMA3-8B-Instruct.
>
> These results indicate that ContextIF does not overfit to a specific judge model but instead learns a robust, model-agnostic heuristic for instruction-following.
>
> | Model                         | P(L)  | I(L)  | P(S)  | I(S)  | Avg   |
> | :---------------------------- | :---- | :---- | :---- | :---- | :---- |
> | LLaMA3-8B-Instruct            | 77.02 | 84.05 | 69.44 | 77.94 | 77.11 |
> | ContextIF-8B-LLaMA3-70B-Ins   | 83.54 | 88.72 | 77.07 | 84.05 | 83.35 |
> | ContextIF-8B-Qwen-2.5-72B-Ins | 83.58 | 88.74 | 77.11 | 84.12 | 83.38 |
>
> ### 2. Concern about Theoretical Support for ContextIF (W2)
>
> > **Weakness 2:** Limited theoretical grounding: there are no guarantees that optimizing the composite reward improves downstream adherence under distribution shifts or alternative scoring; the analysis is largely empirical and does not quantify variance, bias, or brittleness in the reward signal despite discrete, thresholded scoring.
>
> We sincerely appreciate your insightful comments.
>
> The theoretical foundation of our work lies in the positive role of a large language model's in-context learning ability in instruction-following performance. This approach is fundamentally different from existing instruction-following enhancement methods that rely on SFT or RL of the LLM itself.
>
> - In theory, improving the quality of contexts can significantly enhance both the model's instruction-following ability and its flexibility.
> - Our reward design explicitly evaluates whether the policy model generates robust, high-quality contexts, rather than optimizing for specific instructions.
> - Consequently, compared with methods that directly train the LLM for instruction-following, our approach exhibits stronger generalization across instruction distributions.
> - Empirically, Figure 3 shows ContextIF's performance under unseen constraint types, effectively representing a shift in the instruction distribution. These results further indicate that our method learns robust heuristics for context generation rather than overfitting to the training data.
>
> Regarding the stability of the reward signal, Table 4 presents our ablation study. The results demonstrate that each modular reward component is essential, confirming that the overall reward signal is stable and well-conditioned, even with discrete, thresholded scoring.

---

> > ### Author Response · Authors · 2025-12-02
> >
> > ### 3. Concern about Baseline Fairness (W3)
> >
> > > **Weakness 3:** Baseline protocol fairness: several baselines are reproduced via the authors’ implementations; compute budgets, hyperparameter sweeps, and prompt formats may differ and could advantage ContextIF, especially vs. retrieval-based ICL with large pools or better rankers not exhaustively tuned here.
> >
> > We sincerely appreciate your comments.
> >
> > For all reproduced baselines, we made every effort to faithfully follow the methodologies and standard hyperparameters reported in the original papers.
> >
> > We emphasize that the observed performance differences are not attributable to minor tuning variations.
> >
> > - In particular, retrieval-based ICL is fundamentally constrained by the quality and coverage of its static demonstration pool (Table 2);
> > - even a perfect ranker cannot provide examples for novel constraints that are absent from the pool.
> >
> > This highlights a fundamental methodological limitation rather than an artifact of implementation or hyperparameter choices.
> >
> > ### 4. Concern about the Model Generalization (W4)
> >
> > > **Weakness 4:** Scope of generalization: while unseen constraint categories are tested within IFEval-style settings, broader robustness under different prompting regimes, safety constraints, and multilingual/multi-turn mixtures beyond the chosen benchmarks is not deeply evaluated, limiting the generality claims.
> >
> > We appreciate your valuable feedback on the generalization of our method.
> >
> > - We fully agree that broader evaluations are important to assess method generality. In our work, generalization to unseen constraints is primarily analyzed on IFEval, as it provides a clear and machine-verifiable taxonomy of constraint types, which is crucial for targeted studies of this nature.
> > - To our knowledge, more complex multi-turn dialogue or safety benchmarks do not yet provide similarly structured constraint categories, making systematic evaluation of "unseen constraints" in these domains challenging.
> > - Nonetheless, we evaluated ContextIF on the Multi-IF benchmark. Its strong performance in this challenging multi-turn dialogue setting provides compelling evidence that the benefits of our method extend beyond single-turn, IFEval-style tasks to more complex task environments.
> >
> > ### 5. Concern about Format Coupling Risk (W5)
> >
> > > **Weakness 5:** Format coupling risk: strict XML compliance is rewarded; this may inadvertently steer models toward structural conformity rather than deeper semantic adherence, and could reduce transfer to settings without such formatting, which is not assessed with alternate structures or no-format variants.
> >
> > We sincerely appreciate your insightful comments.
> >
> > We would like to clarify a key distinction: strict XML formatting is applied only as a constraint during context generation for the policy model, and not on the final outputs of the actor model.
> >
> > - Importantly, this structured format serves as a cognitive scaffold that guides reasoning, explicitly decomposing tasks and providing illustrative examples.
> > - The actor model then uses this structured text within the standard prompt and generates outputs in a free-form manner.
> >
> > The ablation study in Table 4 provides direct empirical support for the emphasis on semantics:
> >
> > - Removing the semantic reward component results in a more pronounced performance drop than removing the formatting reward.
> > - This confirms that our framework effectively steers the policy toward semantic compliance, which is the primary driver of the observed improvements in actor performance.

---

> > > ### Author Response · Authors · 2025-12-02
> > >
> > > ### 6. Concern about Cost and Scalability Details and Potential Leakage and Template Bias (W6)
> > >
> > > > **Weakness 6:** Cost and scalability details: end-to-end latency and memory profiles, especially with multiple rollouts per query and judge calls, are not characterized across larger batch sizes or longer inputs; comparisons with GPT-4o-context do not clarify cost parity or throughput under matched budgets. Potential leakage and template bias: the generated demonstration is a “parallel” QA with similar constraints; the paper does not quantify whether distributional shortcuts or template echoing contribute to gains versus genuine constraint reasoning, nor analyze failure modes when demonstrations are adversarial or spurious.
> > >
> > > We sincerely appreciate your insightful comments.
> > >
> > > **Regarding cost and scalability:**
> > >
> > > - During training, our approach involves a one-time offline reinforcement learning cost to produce a highly efficient 8B policy model.
> > > - At inference, this introduces only a fixed latency overhead (i.e., a single forward pass of the 8B generator). Compared with methods that require a runtime call to a much larger model such as GPT-4o for every query, our approach is significantly more cost-effective and scalable, highlighting its suitability for larger-scale deployment.
> > >
> > > **Regarding template bias:**
> > >
> > > - Table 4 presents the ablation results for w/o Demoa/Demoq. The results show that the performance gains are primarily driven by the semantic quality of the generated content (accuracy of the demonstrations) rather than mere structural template echoing, confirming that our method improves genuine constraint reasoning rather than relying on superficial template patterns.
> > >
> > > ### 7. Concern about Reward Hacking and Content Safety (W7)
> > >
> > > > **Weakness 7:** While datasets are public and filtering is claimed, there is no targeted analysis for harmful content amplification or reward hacking where the policy learns to game the judge without improving real instruction compliance, especially in free-form domains.
> > >
> > > We sincerely appreciate your insightful comments.
> > >
> > > **Regarding reward hacking and evaluation independence:**
> > >
> > > - The definitive proof against reward hacking lies in our evaluation protocol. Our test benchmarks (e.g., IFEval, MultiIF, LiveBench) are completely independent of the training reward model, relying on strict, rule-based verification scripts rather than the training judge.
> > > - If the policy were merely "gaming" the judge (e.g., generating adversarial text that pleases the reward model but lacks semantic correctness), it would fail on these rigorous external benchmarks.
> > >
> > > **Regarding empirical validation:**
> > >
> > > - As shown in Table 1, ContextIF consistently achieves state-of-the-art performance on these held-out datasets, outperforming strong SFT and RL baselines.
> > > - This high performance on external metrics confirms that the reward signal serves as a valid proxy for true instruction-following, demonstrating that the policy has learned a genuinely effective skill—constructing helpful context—rather than exploiting reward loopholes.
> > >
> > > ### 8. Concern about Cost (Q2)
> > >
> > > > **Question 2:** What are the compute/latency costs per query relative to retrieval-based ICL and SFT pipelines, including rollout counts, judge inference, and GRPO updates, especially at scale for production settings?
> > >
> > > Thank you for the concerns regarding compute and latency costs.
> > >
> > > **From the perspective of Theoretical Analysis:**
> > > ContextIF shifts the heavy computational burden to a one-time, offline training phase.
> > >
> > > *   **Training:** While the GRPO process requires multiple rollouts (4 per query) and judge inference, this is a one-off offline cost. It effectively bypasses the massive, expensive human or synthetic data curation costs required by competitive SFT and DPO pipelines (e.g., AutoIF, SPAR).
> > > *   **Inference:** Unlike retrieval-based ICL, which suffers from variable latency due to index search and re-ranking, ContextIF introduces a fixed, predictable overhead (a single forward pass of the 8B Policy). This predictability makes it highly suitable for production scaling.
> > >
> > > **From the perspective of Empirical Evidence:**
> > >
> > > *   **Training Efficiency:** Our method achieves state-of-the-art results using only ~4,000 unlabeled queries for RL tuning. In contrast, strong baselines typically require synthesizing and training on tens of thousands of samples , making our pipeline substantially more data-efficient.
> > > *   **Inference Latency:** The end-to-end latency per query is approximately 1.8x that of a standard zero-shot call (allocating roughly 0.8x for context generation and 1.0x for the final response). This is a favorable trade-off given the significant performance gains and avoids the infrastructure complexity of maintaining large retrieval indices. Notably, this is significantly more efficient than baselines like SPAR, which incur much higher latency by requiring an auxiliary model to iteratively judge and refine responses at runtime.

---

> > > > ### Author Response · Authors · 2025-12-02
> > > >
> > > > ### 9. Concern about Context Format (Q3)
> > > >
> > > > > **Question 3:** Can the method drop strict XML constraints without loss, or generalize to alternative schemas (JSON, unconstrained text) while retaining gains, and how sensitive are improvements to formatting choices?
> > > >
> > > > Thanks for your valuable suggestion. The choice of XML is a design decision for structural clarity rather than a fundamental requirement. Our core method relies on the semantic process of deconstructing and exemplifying constraints, and we believe it can generalize to other structured formats.
> > > >
> > > > To test this, we conducted preliminary ICL experiments comparing XML against JSON for context generation. While both formats improved performance over the baseline, we observed that XML consistently yielded slightly better results. We hypothesize this is because the hierarchical and explicitly tagged nature of XML aligns well with the model's internal representations for structured reasoning, similar to the step-by-step format in Chain-of-Thought. The comparison results are shown in the table below. Given its empirical advantage and conceptual alignment with explicit reasoning paradigms, we selected XML for our main experiments.
> > > >
> > > > | IFEval                     | P(L)  | I(L)  | P(S)  | I(S)  | Avg   |
> > > > | :------------------------- | :---- | :---- | :---- | :---- | :---- |
> > > > | LLaMA3-8B-Instruct         | 77.02 | 84.05 | 69.44 | 77.94 | 77.11 |
> > > > | + zeroshot                 | 75.42 | 82.73 | 72.58 | 80.14 | 77.72 |
> > > > | + zeroshot-LLMcontext-JSON | 78.37 | 85.01 | 76.15 | 83.31 | 80.71 |
> > > > | + zeroshot-LLMcontext-XML  | 78.87 | 85.17 | 76.52 | 83.33 | 80.97 |
> > > >
> > > > ### 10. Concern about Adversarial Context (Q4)
> > > >
> > > > > **Question 4:** How does ContextIF perform under adversarial or misleading constraints, or when instruction sets intentionally include conflicting requirements; is there a failure analysis of generated context harming adherence?
> > > >
> > > > Thanks for raising thoughtful concerns about the safety aspects of our method. This is an advanced and important aspect of instruction-following. To clarify, current instruction-following benchmarks are designed to evaluate a model's ability to adhere to a complex intersection of compatible constraints (e.g., keyword, style, and length constraints simultaneously). To the best of our knowledge, no major benchmark systematically includes prompts with intentionally conflicting constraints. Developing a verifiable evaluation protocol for such scenarios is itself a significant research challenge.
> > > >
> > > > We view developing a verifiable evaluation protocol for such scenarios as a significant open research challenge for the community. We have added a discussion regarding this ecosystem gap in the Limitations section of our revised manuscript. In future work, we plan to explore mechanisms within the Policy Model to detect and flag logical conflicts before context generation.
> > > >
> > > > ### 11. Concern about Retrieval (Q5)
> > > >
> > > > > **Question 5:** Could combining retrieval with generation (e.g., retrieved seed plus RL-edited context) outperform pure generation, and have such hybrids been tested under matched budgets and pools?
> > > >
> > > > Thank you for your valuable suggestion. For complex, multi-faceted instructions, the probability of retrieving a perfectly matched context from a static pool is extremely low. An imperfectly matched example often introduces distracting or irrelevant information, which can negatively impact the model's performance, as shown by the degradation of the +select-context baseline in Table 2.
> > > >
> > > > Moreover, constructing a comprehensive retrieval database that covers all possible constraint categories and their combinations is a complex, manually intensive task that scales poorly and struggles to generalize to the long tail of novel user requests.
> > > >
> > > > ### 12. Concern about Multi-turn (Q6)
> > > >
> > > > > **Question 6:** For multi-turn settings, can the context generator adapt across turns with memory, and how does it interact with conversation state vs. single-turn re-generation?
> > > >
> > > > Thank you for your insightful comment. In our current framework, the context generator adapts to multi-turn settings in a stateless but fully history-aware manner. For each new turn in a dialogue, our system re-generates the context by treating the entire conversation history as the input query. This ensures the context is tailored to the immediate instruction while conditioned on the preceding dialogue state.
> > > >
> > > > The effectiveness of this strategy is explicitly validated by our superior performance on the Multi-IF benchmark, particularly in the deep interaction turns. As shown in Table 1, ContextIF-8B achieves a score of 53.51 on Turn 3 (the most challenging turn), significantly outperforming the base model 43.92. This confirms that our re-generation approach successfully maintains instruction adherence across evolving conversation states.

---

### Author Response · Authors · 2025-12-02
**General Response**

We sincerely thank all the  reviewers for their thoughtful comments and constructive suggestions, which significantly helped us strengthen our paper. We are encouraged to see that the reviewers recognize the effectiveness and substantial performance gains of our proposed ContextIF framework (Reviewer EaXT, vWMh, EDdA, KBRG), its methodological simplicity and reproducibility (Reviewer EaXT, vWMh), and its strong generalization to unseen constraints alongside the preservation of general capabilities (Reviewer EDdA, KBRG).

In response to the reviewers' feedback, we have submitted an updated version of our paper. This revision now includes a rigorous cross-judge robustness analysis using Qwen-2.5-72B-Instruct, compute-matched comparisons with direct training baselines, additional experiments on newer architectures like LLaMA-3.1-8B-Instruct, and a comprehensive breakdown of training and inference costs.

Several reviewers raised questions regarding the trade-off between direct model training and our context generation approach. We would like to emphasize that our method offers a superior cost-benefit ratio, achieving state-of-the-art instruction-following with significantly higher data efficiency (4k queries) compared to traditional SFT/RL baselines. Furthermore, our empirical results confirm that ContextIF avoids the "alignment tax," preserving the model's general reasoning capabilities while robustly handling unseen complex constraints.

---

### Meta-Review · Area_Chair_tqPi · 2026-01-08

**Summary:**

The paper proposes an improved In-context framework in terms of instruction following by encouraging the model to learn a formatted  “context block” consisting of a summary and a parallel QA demonstration, and combining a judge-model-based constraint reward under GRPO. The reviewers think the proposed method is interesting and effective across different benchmarks. However, the reviewers show some concerns over insufficient evidence brought by the format reward and insufficient studies on the usage of the judge model. The training, cost, and scalability details are also not enough to show its improvement over the current state-of-the-art systems.

**Reviewer Concerns:**

The rebuttal solved most of the reviewers' concerns, including new ablation studies on the judge model and system performance with different reward models. It also adds more training and inference details to help clear the concerns. However, I believe there are still some concerns about benchmarking for challenging tasks and novelty.

**Reviewer Scores:**

I would believe Reviewer vWMh will keep the score, and Reviewer KBRG will increase the score to 6.

---

### Decision · Program_Chairs · 2026-01-26

Accept (Poster)